# ELIMINATING POSITION BIAS OF LANGUAGE MODELS: A MECHANISTIC APPROACH

**Ziqi Wang**[1]    **Hanlin Zhang**[2]    **Xiner Li**[3]    **Kuan-Hao Huang**[1,3]    **Chi Han**[1]

**Shuiwang Ji**[3]    **Sham M. Kakade**[2]    **Hao Peng**[1]    **Heng Ji**[1]

## ABSTRACT

Position bias has proven to be a prevalent issue of modern language models (LMs), where the models prioritize content based on its position within the given context. This bias often leads to unexpected model failures and hurts performance, robustness, and reliability across various applications. A simple mechanistic analysis attributes the position bias to two components employed in nearly all state-of-the-art LMs: causal attention and position embedding. Based on the analyses, we propose to **eliminate** position bias (e.g., different retrieved documents' orders in QA affect performance) with a **training-free zero-shot** approach. Our method changes the causal attention to bidirectional attention between documents and utilizes model attention values to decide the relative orders of documents instead of using the order provided in input prompts, therefore enabling **P**osition-**IN**variant inferenc**E** (**PINE**) at the document level. By eliminating position bias, models achieve better performance and reliability in downstream tasks, including LM-as-a-judge, retrieval-augmented QA, molecule generation, and math reasoning. Notably, PINE is especially useful when adapting LMs for evaluating reasoning pairs: it consistently provides 8 to 10 percentage points performance gains, making `Llama-3-70B-Instruct` perform even better than `GPT-4-0125-preview` and `GPT-4o-2024-08-06` on the RewardBench reasoning set.[1]

## 1 INTRODUCTION

Language models (LMs) (Brown et al., 2020; Chowdhery et al., 2022; Touvron et al., 2023; Achiam et al., 2023) demonstrate impressive performance in general language tasks such as dialogue (Thoppilan et al., 2022), reasoning (Chowdhery et al., 2022), and schema induction Li et al. (2023). However, they tend to favor content at certain positions (Zheng et al., 2024b;a; Wang et al., 2023; Dominguez-Olmedo et al., 2023; Zhu et al., 2023; Chen et al., 2024b; Liu et al., 2024), which harms complex reasoning (Chen et al., 2024b), long-context understanding (Liu et al., 2024) and model-based evaluation (Zheng et al., 2024b). For example, LMs tend to favor the first when it is required to compare the quality of two candidate responses (Zheng et al., 2024b), which hurts their reliability when being used as evaluators (Figure 1 upper); vision-language models perform better in the recognition when the target content is presented at the bottom of the image (Figure 1 lower right, see more examples in Appendix A). Different from *ad hoc* solutions from previous works (Ratner et al., 2023; Cai et al., 2023; Hao et al., 2022; Junqing et al., 2023; Zhu et al., 2023), we seek to understand the causes of position bias and propose to eliminate (not just mitigate) the position bias without training and searching.

We start by analyzing the key components of state-of-the-art LMs – Casual Attention and Position Embedding. They are the key to the success of Transformers (Vaswani et al., 2017), and are also the only two operations in Transformers (Vaswani et al., 2017) that will bring undesirable position bias. This is because other operations do not change representations when position changes (Section 3.2). Moreover, we find an interesting phenomenon through simple experiments and give our hypothesis:

---

[1] [1] University of Illinois Urbana-Champaign, [2] Harvard University, [3] Texas A&M University. Correspondence: ziqiw9@illinois.edu

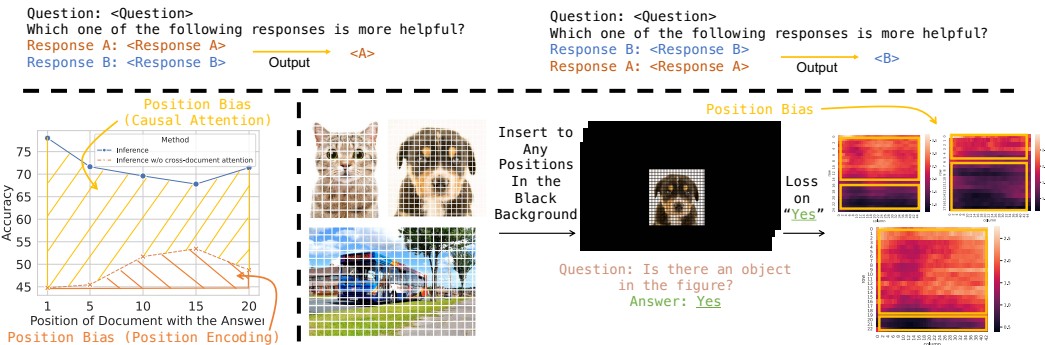

Figure 1: Motivating examples showing how position bias affects model outputs. **Upper**: LMs are asked to select a more helpful one from two given responses. The example shows that LMs are prone to prefer the response positioned at first. **Lower Left**: LMs (`Llama-3-8B-Instruct`) are presented with 20 documents to answer a question, with only one document (the gold-standard document) containing the correct answer. The blue curve represents normal inference, while the red curve represents inference without inter-document attention (RoPE position encodings are kept, a concrete implementation is shown in the middle of Figure 3). The height change of the yellow and orange area reflects the position bias brought by causal attention and RoPE: causal attention generally favors distant content, but RoPE prefers nearby content. **Lower Right**: We insert a real-world image to a large black background image at different positions and prompt VLMs (`Fuyu-8B` (Bavishi et al., 2023)) to compute the loss on the ground truth token. We observe a consistent pattern that models have lower losses (black color) when images are presented at the bottom.

the most popular Rotary Position Embedding (Su et al., 2024) is shown to have recency bias (Su et al., 2024; Peysakhovich & Lerer, 2023) due to its long-form attention weight decay w.r.t. the increase of relative positions, and the causal attention forces unidirectional information propagation, enabling models to pay more attention to distant content. Figure 1 lower left shows this retrieval-augmented QA (Liu et al., 2024) experiment. The height change of the yellow area and orange area reflects the position bias of causal attention and RoPE. Since the yellow area is mostly wider at the beginning and the orange area generally becomes wider at the end (except for the last data point), this shows that the causal attention generally tends to favor distant content, while RoPE generally tends to favor nearby content.[2]

Since attention and position embedding are the causes of position bias, we propose PINE that can eliminate position bias by manipulating causal attention and RoPE to attend to different content equally. Take the retrieval augmented QA (Liu et al., 2024), a task requiring LMs to answer questions based on retrieved documents, for example. The orders of retrieved documents should not affect the final results. To achieve this, we make the inter-document attention bidirectional so that the attention mask will equally attend to all documents. Next, we compute importance scores between documents and use them to re-assign document positions so that positions in the original inputs are discarded. We prove resulting approach enables **P**osition-**in**variant infer**e**nce (PINE) w.r.t. documents in a **training-free/zero-shot** manner.

To justify the effectiveness of PINE, we select four tasks: LM-as-a-judge (Zheng et al., 2024b), which prompts LMs to choose the more helpful one from two given responses to a question; retrieval-augmented question-answering (Liu et al., 2024); molecule generation based on provided properties; and math reasoning. In different tasks, "documents" have different meanings: responses in LM-as-a-judge, properties in molecule generation, and conditions in math reasoning. Notably, we find our method especially useful when LMs are used to assess reasoning pairs: PINE with `Llama-3-70B-Instruct` perform even better than `GPT-4-0125-preview` and `GPT-4o-2024-08-06` on the RewardBench (Lambert et al., 2024a) reasoning set.

To summarize, we:

---

[2]More supporting experiments to this hypothesis in Section 4.3.

- We first revisit the causes of position bias in transformers: causal attention and position encoding (Section 3.2), and then propose a training-free approach dubbed PINE that can eliminate (with proof) the position bias given documents presumed to be position-invariant (Section 3.3).

- Four popular tasks across the general domain to expert domains (chemistry and math) show PINE can bring performance gains consistently across different models and sizes.

## 2 RELATED WORK

**Position Bias in LMs**. Position bias widely exists in LMs (Zheng et al., 2024b;a; Wang et al., 2023; Zhu et al., 2023; Chen et al., 2024b; Liu et al., 2024; Shi et al., 2024). The LM-as-a-judge task offers models two candidate responses to a question and asks models to select the more helpful one. It turns out that LM has a primacy bias that tends to favor the first response (Zheng et al., 2024b). Retrieval-augmented QA asks LM to answer a question based on retrieved documents. Liu et al. (2024); Peysakhovich & Lerer (2023) find that LMs are prone to answer correctly when the document that contains the correct answer is presented at the beginning and the end of retrieved documents. Zheng et al. (2024a) points out that models favor options at certain positions (e.g., prefer "A") in multiple-choice QA. In the in-context learning task, Zhang et al. (2024a); Xu et al. (2024) find that the order of in-context examples affects the final performance. Recently, several papers have proposed to understand the nature of position bias through prompting (Zhang et al., 2024b) and calibration (Hsieh et al., 2024). Our paper analyzes the phenomenon from the mechanical perspective: the computation must be positional-invariant to eliminate position bias.

**Position Bias Solutions in LMs**. *Mitigating* position bias have been studied by many literature from many aspects, such as data augmentation with training (Junqing et al., 2023; Zhu et al., 2023), content sorting by attention value during inference (Peysakhovich & Lerer, 2023), searching (Yu et al., 2024; Adila et al., 2024), calibration (Hsieh et al., 2024), or removing position encoding (Kazemnejad et al., 2024). Moving one step forward, some other solutions are designed to *eliminate* position bias. Zheng et al. (2024a;b) use permutation then average on classification tasks, which will have unacceptable $\mathcal{O}(k!)$ ($k$ is the number of segments) computational overhead when $k$ is large. Hsieh et al. (2024) assumes that the position bias and real relevance are linear combinations and propose solutions accordingly. Different from them, we aim to *eliminate* the position bias from the mechanical perspective without any assumption at a reasonable cost. Although several existing approaches are from the mechanical perspective (Ratner et al., 2023; Cai et al., 2023; Hao et al., 2022), they only perform well in classification tasks and fail in a more general setting: language generation.

## 3 METHODOLOGY

We start by running an example to illustrate position bias, followed by analyzing the cause of position bias, and end with our own approach PINE.

### 3.1 FORMULATION

We take retrieval-augmented QA as an example, where current LMs' performance may greatly suffer from position bias (Liu et al., 2024). The task requires the model to answer a question based on a set of given retrieved documents, where only one of them contains the correct answer. The system prompt SYS is: "`Write a high-quality one-sentence answer for the given question using only the provided search results (some of which might be irrelevant).`". Given a question Q, and three retrieved documents: $\mathcal{D}_1$, $\mathcal{D}_2$, and $\mathcal{D}_3$, we can formulate several different inputs. For example, $[\text{SYS}|\text{Q}|\mathcal{D}_1|\mathcal{D}_2|\mathcal{D}_3]$, and $[\text{SYS}|\text{Q}|\mathcal{D}_2|\mathcal{D}_3|\mathcal{D}_1]$. We expect models to have the same output for these inputs because $\mathcal{D}_2, \mathcal{D}_3, \mathcal{D}_1$ are **position-agnostic documents**: their relative order is not supposed to affect the final result. However, the current LMs answer differently when presented with these different inputs and tend to answer correctly when the document contains the answer at the beginning or at the end of all documents (Liu et al., 2024). The systematic differences of model outputs caused by relative positions of documents reflect the **position bias** of the model. Therefore, current LMs cannot conduct **inter-document position-invariant** inference, and our goal is to make the inference invariant w.r.t. relative document orders. In the rest of this section, we will use this running example. However, we

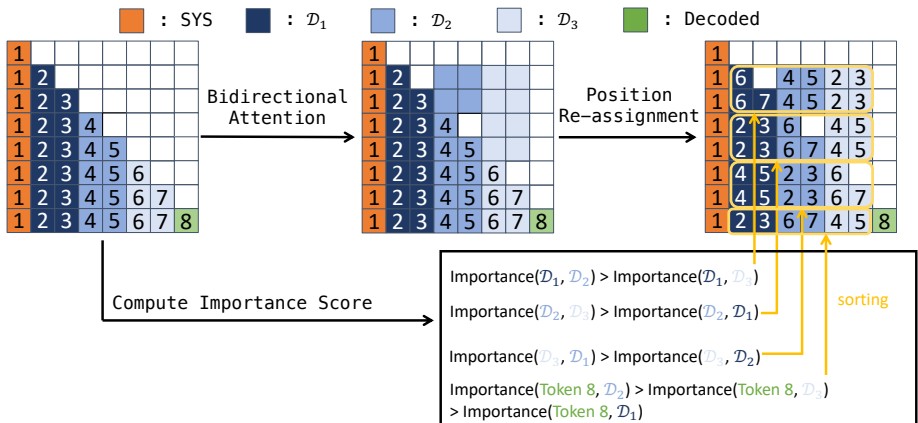

Figure 2: PINE: inter-document position-invariant inference via bidirectional attention. The attention matrix of the running example in Section 3.1 is at the left of the figure, the orange, different blue, and green colors denote system prompts (1 token), three different documents (2 tokens each) and decoded tokens (1 token), respectively. The number at $(i, j)$ in the figure, $p_{ij}$, denotes the position of a token $j$ when computing the attention from query $\mathbf{q}_i$. Therefore, $p_{\cdot j}$ is equal for all $i$ in vanilla inference. PINE enables inter-document bidirectional attention and then uses attention scores between documents to compute their importance. Then, documents are sorted by importances: more important documents are placed in closer positions. The computation of "importance score" is introduced in Section 3.3.

emphasize that "documents" have different meanings in different tasks: responses in LM-as-a-judge, properties in molecule generation, and conditions in math reasoning. Therefore, readers should be aware that the method is not just designed for a single task. In the rest of the paper, we merge SYS and Q into SYS for simplicity.

## 3.2 CAUSAL ATTENTION AND POSITION EMBEDDING ARE THE CAUSE OF POSITION BIAS

Feed-forward networks (FFNs), Query, Key, and Value (QKV) projections, and layer normalization in the Transformer architecture do not cause position bias, as they yield the same representations regardless of document positions. Rather, the attention computation that leads to the position bias:

$$\mathbf{Q}_{\text{PE}} = \text{PE}(\mathbf{Q}, \mathbf{pos}_{\mathbf{Q}}), \mathbf{K}_{\text{PE}} = \text{PE}(\mathbf{K}, \mathbf{pos}_{\mathbf{K}})$$
$$\mathbf{H} = \text{Softmax}\left(\mathbf{Q}_{\text{PE}}\mathbf{K}_{\text{PE}}^T/\sqrt{d}\right) \odot \mathbb{1}_{\text{causal}} \mathbf{V}$$

$$(1)$$

where $\mathbf{Q}, \mathbf{K}, \mathbf{V} \in \mathbb{R}^{n \times d}$ are queries, keys, and values, PE donotes the position encoding, $\mathbf{pos}_{\mathbf{Q}}$ and $\mathbf{pos}_{\mathbf{K}}$ denote the position of queries and keys, and $\mathbb{1}_{\text{causal}}$ denotes the causal attention mask. Eq. 1 reveals that (1) the PE function yields different representations for documents if their orders changes, therefore affecting the importance score $Q_{\text{PE}}K_{\text{PE}}^T$ and hidden states; (2) the $\mathbb{1}_{\text{causal}}$ generates different attention masks for the input documents if we change their positions, resulting in different hidden states. To achieve inter-document position-invariant inference, **H must remain the same regardless of documents' orders**.

## 3.3 PINE: INTER-DOCUMENT POSITION-INVARIANT INFERENCE VIA BIDIRECTIONAL ATTENTION

Our goal is to obtain an inter-document position-invariant hidden state $\mathbf{H}_{\text{PINE}}$, which does not change regardless of document orders. We can mechanistically eliminate the position bias by equally attending to all documents. Therefore, we propose PINE, an approach that uses bidirectional inter-segment attention and re-assigning positions by importance scores (computed from attention score) to eliminate position bias (Figure 2). We address that the "elimination" and "invariance" in our method are talked about from the input-output perspective, i.e., outputs remain unchanged regardless of the

input-position orders. PINE still uses position encoding and does not eliminate position encoding itself.

**Bidirectional Attention.** We first change the attention mask so that documents can attend to each other. Specifically, we make the inter-document attention **bidirectional** but keep the intra-document attention **causal** (Figure 2, middle). Our goal is to eliminate "inter" position bias among different documents rather than "intra" position bias within each document. The latter will lose the order information of tokens, and models can degenerate into bag-of-words models, which is not what we expect.

**Re-assign Positions: Sorting By Importance Scores.** Re-assigning positions must consider two folds: the position of queries and keys. Each token in conventional LMs has the same position when serving as both query and key. In the bidirectional attention we use, this assignment has to be reconsidered. First, LMs are trained causally, meaning the position of the query must be larger than the keys in the attention computation. Therefore, it is necessary to manipulate positions so that each document is the **last** document when serving as queries (the diagonal of the rightmost figure in Figure 2). For tokens before and after documents, their positions are not affected when serving as queries.

Re-assigning positions for keys must be redesigned to eliminate position bias. We determine the positions of documents based on importance scores when they serve as keys (numbers in the rightmost part of Figure 2). Specifically, we first compute the attentions without position embedding involved: $\text{Importance}_{\text{token}}(i,j) = \text{Softmax}(\mathbf{q}_i \mathbf{k}_j^T / \sqrt{d})$, where $d$ is the hidden state dimension. Then, we obtain the importance score between documents by aggregation. For example, $\text{Importance}(\mathcal{D}_1, \mathcal{D}_2) = \sum_{i \in \mathcal{D}_1, j \in \mathcal{D}_2} \text{Importance}_{\text{token}}(i,j) / |\mathcal{D}_2|$. The length normalization is to prevent assigning higher importance scores to longer documents.[3] The importance score could also be computed between individual tokens (e.g., Token 8) and documents. Lastly, we re-assign positions by importance scores as shown in the rightmost part of Figure 2: more important documents will have closer positions to the query. The rightmost part of Figure 2 shows the concrete position re-assignment for keys (its diagonal also represents the position re-assignment for queries). To avoid confusion, we address the fact that we do not actually sort tokens and only re-assign them to different positions. In our position re-assignment, the position of keys may vary depending on the queries (numbers in column are different), which is the key difference between PINE and vanilla inference. Besides, our method is not limited to specific position embedding types.

**Inter-Document Position Invariant Inference.** Once we have new attention mask and position re-assignment, we can place them into Equation 1, and obtain $\mathbf{H}_{\text{PINE}}$. By applying $\mathbf{H}_{\text{PINE}}$ to every layer, attention heads, and tokens, we reach our method PINE. We prove that:

**Lemma 1.** *If the input $\mathbf{Q}, \mathbf{K}, \mathbf{V}$ are inter-document position-invariant representations, then $\mathbf{H}_{PINE}$ are also inter-document position-invariant representations.*

**Proof:** To simplify the notation and without loss of generality (w.l.o.g), we still use examples in Section 3.1.

First, the SYS tokens already satisfy this lemma under the vanilla inference since they appear before documents, and PINE does not change their computation process. We only need to show PINE can make $\mathcal{D}_i$ and Token 8 (i.e., tokens after documents) satisfy the lemma. W.l.o.g, we use $\mathcal{D}_1$ as a running example:

- PINE first obtains importance score between documents: $\text{Sim}(\mathcal{D}_1, \mathcal{D}_i) = \sum \text{Softmax}(\mathbf{Q}_1 \mathbf{K}_i^T / \sqrt{d}) / |\mathcal{D}_i||$, where $\mathbf{Q}_1 \in \mathbb{R}^{2 \times d}, \mathbf{K}_i \in \mathbb{R}^{2 \times d}$, 2 denotes the number of tokens in documents, and $d$ denotes hidden states dimensions. Note that here the $\mathbf{Q}, \mathbf{K}$ have not been applied to position embedding yet. Therefore, the importance score is not a function of input document positions.

- W.l.o.g, let's assume $\text{Sim}(\mathcal{D}_1, \mathcal{D}_2) > \text{Sim}(\mathcal{D}_1, \mathcal{D}_3)$, then we sort the document as follows $[\mathcal{D}_3 | \mathcal{D}_2 | \mathcal{D}_1]$ when they serve as keys and $\mathcal{D}_1$ as query. Concretely, $\mathbf{Q}_{\text{PE},1} = \text{PE}(\mathbf{Q}_1, 3)$ (3 denotes it is treated as the last, i.e., third, document), $\mathbf{K}_{\text{PE},1} = \text{PE}(\mathbf{K}_1, 3)$, $\mathbf{K}_{\text{PE},2} = \text{PE}(\mathbf{K}_2, 2)$,

---

[3]In our pilot experiments, we find summation makes models convert from position bias to length bias. We also try maximum instead of averaging and find this methods usually have noticeably worse performance than averaging possibly due to noises brought by unimportant tokens.

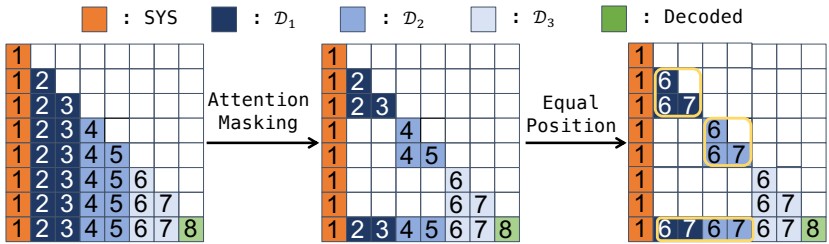

Figure 3: Previous work PCW (Ratner et al., 2023) eliminates position bias by first masking all inter-document attention and then re-assigning all documents the same positions. The notions are kept the same as Figure 2. Our experiment in Section 4 shows that PCW brings severe performance drop for tasks requiring language generation.

$\mathbf{K}_{\text{PE},3} = \text{PE}(\mathbf{K}_3, 1)$. Then we compute hidden states of $\mathcal{D}_1$: $\mathbf{H}_1 = \text{Softmax}(\mathbf{Q}_{\text{PE},1}\mathbf{K}_{\text{PE}}/\sqrt{d})$, where $\mathbf{K}_{\text{PE}}$ is the key values for the whole sequence $[\text{SYS}|\mathcal{D}_3|\mathcal{D}_2|\mathcal{D}_1]$. It is noted that this process does not use any variables that are dependent on the input document positions, nor directly use the input document positions. Therefore, $\mathbf{H}_1$ obtained by PINE is not a function of input document positions.

- Similarly, $\mathbf{H}_2$, $\mathbf{H}_3$, and Token 8's hidden states are not functions of input document positions. Their concatenation yields $\mathbf{H}_{\text{PINE}}$, which is not a function of input document positions.

**Proof ends.**

**Theorem 1.** *Given an input, if $\mathbf{H}_{PINE}$ is applied to every layer, attention head, and token to replace the conventional attention computation, then the model outputs are inter-document position-invariant representations.*

The theorem can be proved by mathematical induction by (1) lemma, (2) FFN, QKV projection, and layer norm yield representations that are not a function of document positions, and (3) the embedding representation is not a function of document positions. We put the complete proof in Appendix B. This lemma can also be understood in a simpler way: it is a corollary of the symmetry principle (Gross, 1996).

Some takeaways that are worth noting: (1) Both bidirectional attention mask and position re-assignment are needed to complete the proof. (2) PINE needs to be applied to every layer, attention heads, and tokens to complete the proof. (3) PINE is not limited to specific position embedding types.

## 3.4 DISCUSSION

**Different Position Re-Assignment Methods.** PINE puts documents with higher importance scores to a closer position to queries. Another option is to put documents with higher importance scores in a more distant position to the queries. Considering the recency bias brought by the most popular rotary position embedding (RoPE) (Su et al., 2024), this alternative approach makes RoPE "disrespect" the attention of models. Therefore, we believe this alternative choice is not optimal, which is justified by our experiments in Section 4.3.

**Different Attention Masks.** Previous work PCW (Ratner et al., 2023) adopts a different way: it masks the inter-document attention instead of making it bidirectional (Figure 3, middle and right). Accordingly, it adopts a simplified position re-assignment method of ours: putting all documents in the same positions. However, masking all inter-document attention loses contextual information (the white part surrounded by colored blocks in Figure 3). Moreover, some different tokens now share the same positions (Figure 3, right), which could confuse models. As a result, PCW performs poorly in language generation tasks (Section 4).

**Inference Cost.** PINE incurrs additional computation overhead due to extra operations. Practically, the extra big $\mathcal{O}$ computation complexity to obtain hidden states is $\mathcal{O}(nk\log k)$, where $n$ and $k$ denote text length and the number of input documents, respectively. The bidirectional attention does not bring extra cost, the position re-assignment brings $\mathcal{O}(k\log k)$ for each token since the sorting

algorithms are involved. The real computation cost is acceptable since $k$ is usually small (e.g., $k = 2$ in the LLM-as-a-judge task and $k = 20$ in the retrieval-augmented QA). Section 4.5 shows results of real-world wall time and memory cost.

## 4 EXPERIMENT

Our experiments aim to show PINE can improve model performance across diverse tasks and have superior performance than other approaches.

### 4.1 SETTINGS

We select four tasks that pose position bias: LM-as-a-judge (Zheng et al., 2024b) that prompts LMs to select a better response out of two given a question, retrieval-augmented question-answering (Liu et al., 2024) that asks LMs to answer questions based on retrieved documents, molecule generation based on provided properties (Ramakrishnan et al., 2014), and math reasoning based on several given conditions Chen et al. (2024b). We follow previous work (Liu et al., 2024; Lambert et al., 2024a) and use temperature 0 in avoid variance.

**LM-as-a-judge.** We benchmark our method on 23 datasets in the RewardBench[4] (Lambert et al., 2024b) that can be categorized into four types: Chat, Chat-Hard, Safety, and Reasoning. We use the official data split, prompts, and evaluation scripts to ensure reproducibility. We use LLaMa-3-Instruct models (AI, 2024) and Qwen-1.5-Chat models (Bai et al., 2023) for experiments. To show how positions affect results, we present four results: the ground-truth response is positioned at first, second, or shuffled, and PINE results (which yield the same results for all three scenarios above).

**Retrieval-augmented QA.** We follow the settings and use the prompts, data, and evaluation scripts of (Liu et al., 2024)[5]: Only one of the retrieved documents (10 or 20 in total) contains the ground-truth answer for the given question. We list prompts in Appendix E. We use LLaMa-3-70B-Instruct model (AI, 2024) for experiment. To show how positions affect results, we present several results: the ground-truth document is positioned at the beginning, middle, last, or shuffled, and PINE results (which yield the same results for all scenarios above).

**Molecule Generation and Math reasoning**. We also conduct two bonus experiments. Molecule generation based on given properties (property positions can be swapped), and math reasoning where conditions can be swapped.

More details of the four tasks can be found in Appendix E. Qualitative examples of the four tasks can be found in Appendix F.

**Baselines.** The goal of PINE is to eliminate position bias during inference mechanically. Therefore, we choose methods that have the same design principle as our baselines: (1) Vanilla inference (2) Vanilla inference with no inter-document attention (NIA for short, i.e., the middle figure in Figure. 3): The latter documents will have no attention to formers. (3) Parallel Context Window (PCW, rightmost in Figure. 3) (Ratner et al., 2023): PCW extends the baseline (2) by manipulating positions of documents. PCW allows all documents to share the same positions. (4) Structured Prompting (SP, a variant version of PCW) Hao et al. (2022): SP extends (3) by lowering attentions between decoded tokens and input documents to $\frac{1}{N}$ to solve the perplexity exploding problem in PCW. Similar to the proof in Section 3.3, we can know that (1) and (2) are not inter-document position invariant, whereas (3) and (4) are. Beyond these methods, we also introduce two other debiasing baselines: permutation (Zheng et al., 2024a) and calibration (Zhao et al., 2021).

### 4.2 RESULTS ON LM-AS-A-JUDGE

**Position bias exists across different models and sizes.** We first analyze the statistics of position bias in RewardBench with different models (Appendix C). Position bias is quite common in RewardBench, and can be up to $48.0\%$. Larger models have less position bias, however, the position bias could still on average affect up to $10\%$ data.

---

[4]Apache-2.0 license. https://github.com/allenai/reward-bench
[5]MIT license. https://github.com/nelson-liu/lost-in-the-middle

Table 1: Main results of RewardBench. Vanilla denotes the normal inference, (GT at A) means the ground truth chosen response is presented at the first, and (GT at B) indicates the second. For the 72B model, we additionally benchmark the Qwen 2.5 model. PINE consistently improves LM's performance across different models and sizes and is particularly useful when assessing reasoning pairs.

| Method | Llama-3-Instruct | | Qwen-1.5-Chat | | | | | |
| | 8B | 70B | 1.8B | 4B | 7B | 32B | 72B / 72B (Qwen 2.5) | 110B |
|---|---|---|---|---|---|---|---|---|
| | | | | RewardBench (Full set) | | | | |
| Vanilla (GT at A) | 67.5 | 78.0 | 36.3 | 29.5 | 61.4 | 74.2 | 79.6 / 87.2 | 87.2 |
| Vanilla (GT at B) | 66.3 | 76.5 | 66.2 | 76.6 | 59.6 | 74.8 | 69.5 / 80.5 | 75.7 |
| Vanilla (Shuffle) | 64.8 | 76.0 | 50.3 | 53.1 | 60.9 | 72.8 | 72.8 / 83.4 | 81.1 |
| PINE | $66.7_{+1.9}$ | $77.4_{+1.4}$ | $52.9_{+2.6}$ | $58.2_{+5.1}$ | $61.5_{+0.6}$ | $74.8_{+2.0}$ | $71.8_{-1.1}$ / $84.5_{+1.1}$ | $82.9_{+1.7}$ |
| | | | | RewardBench (Reasoning set) | | | | |
| Vanilla (GT at A) | 80.3 | 87.8 | 43.3 | 42.8 | 62.1 | 78.3 | 83.0 / 93.7 | 90.0 |
| Vanilla (GT at B) | 66.0 | 80.3 | 57.2 | 62.3 | 54.3 | 73.6 | 68.7 / 76.0 | 73.0 |
| Vanilla (Shuffle) | 65.3 | 78.9 | 48.4 | 54.1 | 59.3 | 66.8 | 68.2 / 85.5 | 78.0 |
| PINE | $73.4_{+8.1}$ | $87.6_{+8.7}$ | $60.1_{+11.7}$ | $61.0_{+6.9}$ | $63.0_{+3.7}$ | $76.7_{+9.9}$ | $69.0_{+0.8}$ / $91.3_{+5.8}$ | $86.2_{+8.2}$ |

Table 2: Baseline performance on RewardBench. PINE achieves superior performance to baseline models, performing $4.8\%$ and $4.7\%$ better than the best performed baseline on two models.

| Method | LLaMa-3-8B-Instruct | | Qwen1.5-7B-Chat | |
| | Reasoning | Full Set | Reasoning | Full Set |
|---|---|---|---|---|
| NIA (GT at A) | 43.7 | 56.3 | 60.7 | 61.3 |
| NIA (GT at B) | 66.7 | 65.8 | 44.1 | 52.2 |
| NIA | 55.9 | 61.9 | 51.4 | 56.8 |
| PCW | 56.5 | 61.7 | 53.4 | 55.2 |
| SP | 55.4 | 60.8 | 52.4 | 55.4 |
| PINE | $73.4_{+16.9}$ | $66.7_{+4.8}$ | $63.0_{+9.6}$ | $61.5_{+4.7}$ |

**PINE consistantly improve model performance across models and sizes.** Table 1 shows the main results on RewardBench. We experiment with Llama-3 and Qwen-1.5 across different model sizes. The position of the ground truth chosen option is randomly shuffled. Therefore, the accuracy of the random guess method is expected to be $50\%$. First, the first two rows reveal that larger models tend to have a primacy bias, whereas smaller models tend to have a recency bias. By comparing the last two rows of each model size, we conclude that models across different sizes perform better with the help of PINE by eliminating position bias. The only exception is the Qwen-1.5-72B-Chat model. We suspect this model is not well-trained since Qwen-1.5-32B-Chat performs the same as the 72B model in vanilla inference, despite half of the model size. Qwen 2 report (Yang et al., 2024) also shows that the Qwen 1.5B 72B model performs even worse than 32B in reasoning. Moreover, Table 1 shows that Qwen 2.5 72B can obtain consistent performance gains. Overall, PINE improves performance from a statistical perspective and makes models more reliable when as evaluators. Full results are shown in Appendix D.

**PINE is extremely useful when assessing reasoning problems in RewardBench.** PINE consistently improves model performance on the "reasoning" subset by a large margin: from 8 to 10 percentage points in most cases. Specifically, LlaMa-3 Instruct 70B was originally ranked 22nd generative model in the reasoning subset of RewardBench. With PINE, it achieves the 7th rank ($87.6\%$), **out-performing `GPT-4-0125-preview` (the previous 8th rank,** $86.9\%$**), `GPT-4o-2024-08-06` (the previous 9th rank,** $86.6\%$**), and `Llama-3.1-405B-Instruct-Turbo` (the previous 7th rank,** $87.1\%$**).**[6].

**PINE performs better than baseline models that adopt different attention masks.** We then compare PINE with baseline models mentioned in Section 4.1 on Llama-3-8B-Instruct and Qwen1.5-7B-Chat model. They adopt a different attention mask: masking inter-document attention instead of making them bi-directional. Since NIA is not inter-document position-invariant, we also apply NIA with two extreme cases: the ground truth chosen response is always in the first or second

---

[6]Results are provided by the official leaderboard (as of Sep 17, 2024): `https://huggingface.co/spaces/allenai/reward-bench`

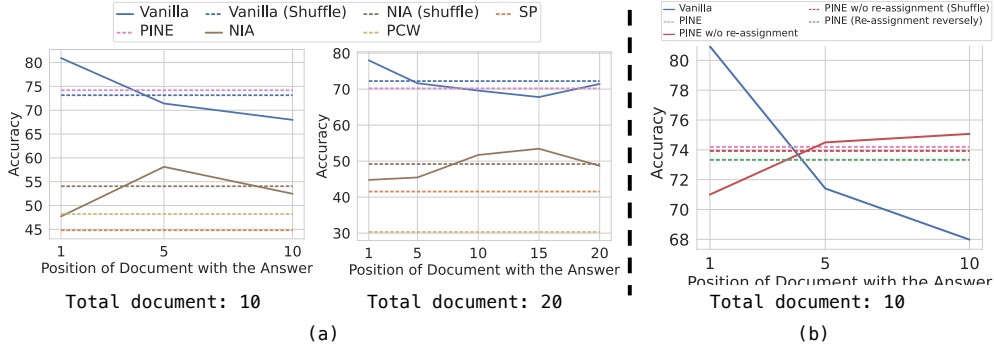

Figure 4: The results of retrieval-augmented QA on Llama-3-70B-Instruct. Dashed lines indicate that the method is either inter-document position-invariant or the result is obtained on the order-shuffled data (denoted in the legend). (a) shows results of PINE against baselines. (b) shows results of different designs of PINE.

place. Results on Table 2 show that PINE achieves the best performance and largely outperforms the best baselines by $\sim 5\%$, and outperforms NIA even if NIA is placed in the extreme case. On the reasoning subset, this performance gap becomes much even greater. The results reveal that masking inter-document attention mask is much less effective than bidirectional inter-document attention mask applied in PINE.

**PINE performs better than permutation and calibration methods.** Another two widely used debiasing methods are permutation (Zheng et al., 2024a) and calibration (Zhao et al., 2021). They are usually used in the logit-based evaluation or single-token generation. Their effectiveness in the open-ended generation is less explored. In our experiments, we find calibration methods generates rubbish responses, which we believe is because of the strong assumption in (Zhao et al., 2021): uniform distribution of all tokens in the generation task. For the permutation methods, we find LLama-3-8B-Instruct have $69.0\%$ and $65.9\%$ accuracy, Qwen1.5-7B-Chat has $58.2\%$ and $61.3\%$ accuracy on the reasoning set and fullest respectively, all underperforming PINE (numbers reported in Table 2).

## 4.3 RESULTS ON RETRIEVAL-AUGMENTED QUESTION-ANSWERING

**PINE performs better than baselines, on-par with vanilla inference on average while not being affected by the worst case.** Models tend to perform better when the gold-standard document is at the beginning and the end of all documents in retrieval-augmented question-answers. Figure 4 (a) shows the results on LLaMa-3-70B-Instruct when 10 or 20 documents were presented. First, it is easy to conclude that all baselines are much worse than PINE (the pink line), which is consistent to the previous experiment. Second, PINE achieves on-par performance on average compared with vanilla inference while being inter-document position invariant. Specifically, PINE is slightly better/worse than vanilla inference with the gap +1.2/-2.0 when there are 10 and 20 documents in total. We hypothesize that the slight performance drop of PINE for the 20 document setting is due to the performance drop of document importance score computation in PINE when presented with many documents. However, PINE is position-invariant, therefore does not be affected by the worst case (the bottom of blue solid curves). Third, the height generally becomes smaller between blue and brown solid lines in Figure 4 (a), and between the blue and red solid lines in Figure 4 (b) when the gold-standard document position increases, reflecting the causal attention generally prefers distant content, which is consistent to the hypothesis in Section 1. The brown line in Figure 4 (a) and red line (b) generally reflect recency bias brought by RoPE, which is consistent to previous works (Su et al., 2024; Peysakhovich & Lerer, 2023).

**PINE performs better than other position assignment methods.** So far, our experiments show that bidirectional inter-document attention is the better design choice than the masked one. However, there are still several design options for the position assignment, as discussed in Section 3.4. The first option is to re-assign position reversely, and the other is to use PINE without position re-assignment (i.e., use input document positions when they serve as keys). To gain a deeper understanding, we extend the

Table 3: The result of molecule generation on QM9 dataset. PINE improves model performance in 5 out of 6 criteria.

| Model | $\alpha$ | $\epsilon_{\text{HOMO}}$ | $\epsilon_{\text{LUMO}}$ | $\Delta\epsilon$ | $\mu$ | $C_v$ |
|---|---|---|---|---|---|---|
| **LLama** | 6.3997 | 103.93 | 53.4 | 99.13 | **3.4112** | 4.3785 |
| **Llama + PINE** | **6.3702** | **102.15** | **53.09** | **98.27** | 3.4917 | **4.2886** |

retrieval-augmented QA experiments with the two mentioned alternative position assignment methods, and the results are presented in Figure 4 (b). The figure tells us that PINE is slightly better than PINE without position re-assignment on average (+0.3. The gap becomes larger when 20 documents are presented: +1.5). Position re-assignment reversely has relatively worse results, showing that PINE is a better design choice, which is consistent with the intuitive analysis mentioned in Section 3.4. Although position re-assignment seems only to bring less gains than bidirectional attention mask, it is required to complete the proof that PINE can *eliminate* the position bias. Therefore, PINE without position re-assignment may suffice if one does not aim to eliminate the position bias and cares more about efficiency (no extra $\mathcal{O}(nk\log k)$ sorting cost).

## 4.4 RESULTS ON MOLECULE GENERATION AND MATH REASONING

**PINE improves model performance on 5 out of 6 criteria in molecule generation .** Table 3 shows the results of molecule generation. The consistent gain in 5 out 6 criteria shows the effectiveness of PINE.

**PINE improves math reasoning capabilities.** Figure 5 shows the results of Qwen1.5 models on R-GSM dataset. It can be shown that PINE outperforms vanilla inference for both small 7B models and large 110B models.

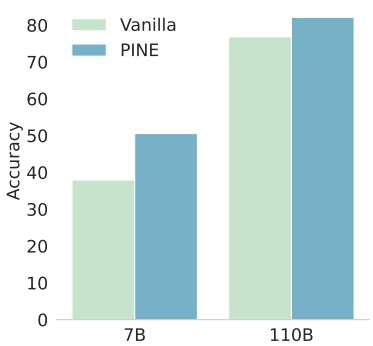

## 4.5 COMPUTATIONAL OVERHEAD

Section 3.4 briefly discusses the computational overhead, with a conclusion that PINE 's efficiency is still doable. In our experiments, we find the wall time of PINE is ∼2x and ∼8x of the vanilla inference on the LM-as-a-judge task and retrieval-augmented QA task with 20 documents, which is acceptable at least during experiments. However, we did not specially optimize codes to accelerate PINE, and our implementation still contains a "for" loop. Therefore, we believe there is room to accelerate PINE. Compared with the time overhead, the memory overhead is small and

Figure 5: Math reasoning results of Qwen1.5 series on R-GSM subset. PINE improves the reasoning accuracy by 12.6% and 5.3% with 7B and 110B models respectively compared with vanilla inference.

PINE can be run with 70B models on 3x A100 80G on the retrieval-augmented QA task, which requires the same number of GPUs as the vanilla inference. Since efficiency is not the main focus of this paper, we leave this as our future work.

## 5 CONCLUSION, LIMITATIONS AND FUTURE WORK

We propose a novel train-free zero-shot approach to eliminate the position bias mechanically. The core idea is to make every input documents equally affected by the attention mask and position embedding. However, PINE requires extra computation. We believe there is room to improve the efficiency with more efficient implementation, and we leave this as our future work.

## REPRODUCIBILITY STATEMENT

Experiment details are described in Section 4.1 and Appendix E. Codes are uploaded to: `https://github.com/wzq016/PINE`. A complete proof of the lemma and theorem occurred in Section 3.3 are presented in Section 3.3 and Appendix B.

## ACKNOWLEDGMENT

We thank Chujie Zheng for the helpful discussion and feedback. This research is based upon work supported DARPA ITM Program No. FA8650-23-C-7316. The views and conclusions contained herein are those of the authors and should not be interpreted as necessarily representing the official policies, either expressed or implied, of the U.S. Government. The U.S. Government is authorized to reproduce and distribute reprints for governmental purposes notwithstanding any copyright annotation therein.

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

## A   ANOTHER EXAMPLE OF POSITION BIAS IN VLMS

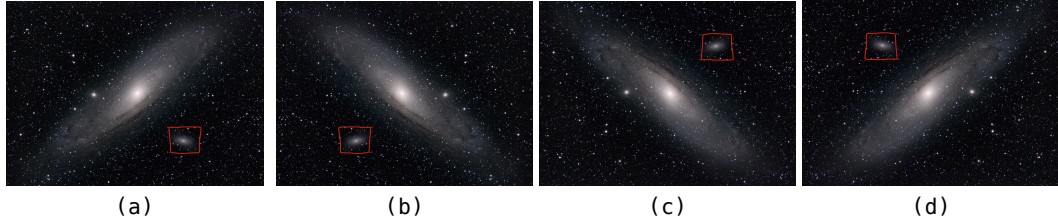

Figure 6: Image of Andromeda Galaxy. GPT-4V is asked to answer the galaxy of the redbox (M110). (b), (c), (d) images are generated by flipping (a). Results show that GPT-4V can answer correctly on (c) and (d) and incorrectly on (a) and (b), revealing position bias in VLMs.

To show VLMs suffer from position bias, we ask models to identify a small object appeared in the full image. We ask GPT-4V (OpenAI, 2023) to identify one satellite galaxy M110 of Andromeda Galaxy (M31). The central objective in the Figure 6 (a) is the Andromeda Galaxy and the galaxy in the red box is M110 satellite galaxy. Another satellite galaxy is M32 which is the brightest part near the Andromeda Galaxy.

We flip the image (a) and obtains images (b), (c), and (d). We prompt GPT-4V "`What galaxy is in the red box?`", and here are the responses we collected:

**(a)**: `The galaxy within the red box is `M32`, also known as Messier 32. It's a dwarf elliptical galaxy and is a satellite of the Andromeda Galaxy (M31), which is the prominent galaxy in the image.  M32 is one of the more well-studied dwarf elliptical galaxies, largely due to its proximity to our own galaxy and its interactions with the Andromeda Galaxy.`

**(b)**: `    The object inside the red box in the image appears to be a smaller galaxy located near the larger, more prominent galaxy (Andromeda Galaxy, M31).  This smaller galaxy is likely `M32`, a satellite galaxy of the Andromeda Galaxy.  M32 is a compact elliptical galaxy and is often visible in images as a smaller, dense, bright spot near Andromeda, as seen in the boxed area of your image.`

**(c)**: `    The galaxy within the red box is `M110`, also known as NGC 205.  M110 is a dwarf elliptical galaxy, which, like M32, is a satellite of the Andromeda Galaxy (M31).  It is one of the members of the Local Group of galaxies, which includes the Milky Way, the Andromeda Galaxy, and about 54 other known galaxies.`

**(d)**: `    The galaxy in the red box appears to be a satellite galaxy of the Andromeda Galaxy, which is the prominent galaxy visible in the rest of the image.  This smaller galaxy is likely `M110` (NGC 205), which is a dwarf elliptical galaxy and a companion to the Andromeda Galaxy, M31.  It's one of the several satellite galaxies gravitationally bound to Andromeda, visible here as a faint, elongated object in the outlined area.`

We can find that models answer corrected when M110 is at the top of the image, revealing that VLMs also suffer from the position bias. The position bias may lead unreliable VLMs when fine-grained image analysis are needed (e.g., small object detection (Wu & Xie, 2023)).

## B   COMPLETE PROOF

This section provided a complete proof to show PINE can eliminate position bias.

To simplify the notation and without loss of generality (w.l.o.g), we still use examples in Section 3.1.

**Theorem 1.** *Given an input, if* $\mathbf{H}_{PINE}$ *is applied to every layer, attention head, and token to replace the conventional attention computation, then the model outputs are inter-document position-invariant representations.*

First, the embedding layer is not a function of input documents positions. Suppose that the $i$th layer's input hidden states are not a function of input documents positions, then within each layer:

- The attention hidden states are not a function of input documents positions (Lemma).

- The Layernorm, FFN outputs are not a function of input documents positions.

- Therefore, the output hidden states of $i$th transformer layer, i.e., the input hidden states of $i + 1$th transformer layer, are not a function of input documents positions.

Using mathematical induction, we know the final outputs are not a function of input documents positions.

**Proof ends.**

Notes on the proof:

- PINE needs to be applied on each layer, attention heads, and tokens to satisfy the above proof.

- The extra big $\mathcal{O}$ computation cost is purely come from the position re-assignment step: $\mathcal{O}(klogk)$ for sorting $k$ documents. Since we need to repeat this step for every token, the extra computation cost is $\mathcal{O}(nklogk)$, where $n$ is the number of tokens.

- Although position re-assignment brings an extra computational cost, it is a must to complete the proof. Removing this step will make PINE unable to "eliminate" position bias. Similarly, a bidirectional attention mask is also a must to complete the proof.

- PINE is not limited to specific position encoding algorithms.

## C   STATISTICS OF POSITION BIAS IN REWARDBENCH

We show the statistics of position bias in RewardBench in Table 4.

Table 4: The portion of data (%) that models have position bias in RewardBench, i.e., models change answers after swaping candidate responses orders. We color the subsets that have more than 25% data causing position bias with cyan.

| Model | Size | Chat | Chat-Hard | Safety | Reasoning | Avg. |
|---|---|---|---|---|---|---|
| LLaMa-3 | 8B | 10.3 | 21.5 | 11.4 | 27.6 | 17.7 |
| -Instruct | 70B | 3.6 | 16.0 | 5.8 | 15.2 | 10.2 |
| | 1.8B | 33.5 | 37.9 | 24.7 | 13.3 | 27.4 |
| | 4B | 48.0 | 38.6 | 57.4 | 12.7 | 39.2 |
| Qwen-1.5 | 7B | 17.0 | 20.6 | 10.9 | 26.5 | 18.8 |
| -Chat | 32B | 7.8 | 20.0 | 9.6 | 26.4 | 16.0 |
| | 72B | 10.9 | 22.6 | 9.6 | 24.7 | 17.0 |
| | 110B | 8.7 | 16.0 | 11.5 | 23.5 | 14.9 |

## D   FULL RESULTS OF REWARDBENCH

Table 5, 6, 7, 8, 9, 10, 11, 12, 13, 14, 15 present the full results of the reward bench. After inspecting the error cases, we categorize the performance drop in the Chat-Hard and Safety subsets into two main aspects:

- The instruction-following capabilities become a bit worse. For example, LLMs tend to solve the "user question" instead of comparing two responses, or LLMs do not output answers in requested formats, causing parsing failures when computing performance scores.

- LMs overly focus on helpfulness in safety prompts, therefore causing performance degradation in the Safety dataset.

However, the positive effect of PINE (i.e., eliminating position bias) is more significant than these negative effects; therefore, the overall PINE is still beneficial to models.

Table 5: Full results of Table 1. Vanilla denotes the normal inference, (GT at A) means the ground truth chosen response is presented at the first, and (GT at B) indicates at the second. PINE consistently improves LM's performance across different model sizes. Consistent to Table 4, we color the subsets with severe position bias cyan. It can be observed that PINE generally improves performance on cyan subsets by a large margin, which is consistent to our motivation and goal.

| Model | Size | Method | Chat | Chat-Hard | Safety | Reasoning | Avg. |
|---|---|---|---|---|---|---|---|
| LLaMa-3 -Instruct | 8B | Vanilla (GT at A) | 90.1 | 35.2 | 64.6 | 80.3 | 67.5 |
| | | Vanilla (GT at B) | 85.3 | 48.7 | 65.3 | 66.0 | 66.3 |
| | | Vanilla | 85.3 | 41.6 | 67.0 | 65.3 | 64.8 |
| | | PINE | 85.6 | 41.5 | 66.5 | 73.4 | $66.7_{+1.9}$ |
| | 70B | Vanilla (GT at A) | 98.6 | 52.0 | 73.6 | 87.8 | 78.0 |
| | | Vanilla (GT at B) | 93.9 | 62.1 | 69.8 | 80.3 | 76.5 |
| | | Vanilla | 97.4 | 58.3 | 69.6 | 78.9 | 76.0 |
| | | PINE | 96.9 | 57.4 | 67.7 | 87.6 | $77.4_{+1.4}$ |
| Qwen-1.5 -Chat | 1.8B | Vanilla (GT at A) | 31.7 | 30.0 | 40.3 | 43.3 | 36.3 |
| | | Vanilla (GT at B) | 69.4 | 72.6 | 65.7 | 57.2 | 66.2 |
| | | Vanilla | 49.7 | 50.9 | 52.0 | 48.4 | 50.3 |
| | | PINE | 30.0 | 59.9 | 61.4 | 60.1 | $52.9_{+2.6}$ |
| | 4B | Vanilla (GT at A) | 32.8 | 24.8 | 17.4 | 42.8 | 29.5 |
| | | Vanilla (GT at B) | 86.6 | 74.5 | 82.9 | 62.3 | 76.6 |
| | | Vanilla | 58.9 | 48.7 | 50.9 | 54.1 | 53.1 |
| | | PINE | 73.0 | 45.2 | 53.7 | 61.0 | $58.2_{+5.1}$ |
| | 7B | Vanilla (GT at A) | 85.5 | 35.9 | 62.4 | 62.1 | 61.4 |
| | | Vanilla (GT at B) | 77.1 | 47.4 | 59.5 | 54.3 | 59.6 |
| | | Vanilla | 77.5 | 44.2 | 62.6 | 59.3 | 60.9 |
| | | PINE | 85.8 | 38.7 | 58.6 | 63.0 | $61.5_{+0.6}$ |
| | 32B | Vanilla (GT at A) | 93.6 | 47.7 | 77.1 | 78.3 | 74.2 |
| | | Vanilla (GT at B) | 91.9 | 52.2 | 81.6 | 73.6 | 74.8 |
| | | Vanilla | 92.7 | 51.2 | 80.5 | 66.8 | 72.8 |
| | | PINE | 93.0 | 49.8 | 79.7 | 76.7 | $74.8_{+2.0}$ |
| | 72B | Vanilla (GT at A) | 95.7 | 59.0 | 80.8 | 83.0 | 79.6 |
| | | Vanilla (GT at B) | 89.0 | 46.5 | 73.7 | 68.7 | 69.5 |
| | | Vanilla | 94.0 | 51.4 | 77.8 | 68.2 | **72.8** |
| | | PINE | 93.9 | 46.1 | 78.2 | 69.0 | $71.8_{-1.1}$ |
| | 72B (Qwen 2.5) | Vanilla (GT at A) | 97.5 | 71.9 | 85.7 | 93.7 | 87.2 |
| | | Vanilla (GT at B) | 95.0 | 67.5 | 83.4 | 76.0 | 80.5 |
| | | Vanilla | 96.6 | 68.0 | 83.3 | 85.5 | 83.4 |
| | | PINE | 96.6 | 67.1 | 83.0 | 91.3 | $74.5_{+1.1}$ |
| | 110B | Vanilla (GT at A) | 98.6 | 70.5 | 89.6 | 90.0 | 87.2 |
| | | Vanilla (GT at B) | 91.1 | 59.2 | 79.5 | 73.0 | 75.7 |
| | | Vanilla | 96.2 | 66.7 | 83.7 | 78.0 | 81.1 |
| | | PINE | 95.5 | 64.8 | 85.0 | 86.2 | $82.9_{+1.7}$ |

Table 6: Full version of Table 2. PINE achieves superior performance to baseline models, performing 4.8% and 4.7% better than the best performed baseline on two models.

| Model | Method | Chat | Chat-Hard | Safety | Reasoning | Avg. |
|---|---|---|---|---|---|---|
| | NIA (GT at A) | 81.0 | 40.7 | 59.7 | 43.7 | 56.3 |
| | NIA (GT at B) | 81.0 | 49.7 | 65.8 | 66.7 | 65.8 |
| LLaMa-3 | NIA | 80.9 | 46.7 | 64.0 | 55.9 | 61.9 |
| 8B-Instruct | PCW | 78.6 | 46.8 | 64.8 | 56.5 | 61.7 |
| | SP | 79.6 | 43.3 | 65.0 | 55.4 | 60.8 |
| | PINE | 85.6 | 41.5 | 66.5 | 73.4 | **66.7**$_{+4.8}$ |
| | NIA (GT at A) | 67.7 | 57.2 | 59.6 | 60.7 | 61.3 |
| | NIA (GT at B) | 67.9 | 35.9 | 61.0 | 44.1 | 52.2 |
| Qwen-1.5 | NIA | 74.9 | 43.5 | 57.4 | 51.4 | 56.8 |
| 7B-Chat | PCW | 67.2 | 42.0 | 58.3 | 53.4 | 55.2 |
| | SP | 69.4 | 41.8 | 58.0 | 52.4 | 55.4 |
| | PINE | 85.8 | 38.7 | 58.6 | 63.0 | **61.5**$_{+4.7}$ |

Table 7: Meta-Llama-3-8B-Instruct results on RewardBench

| Dataset | Vanilla | PINE |
|---|---|---|
| alpacaeval-easy | **91.0** | 90.0 |
| alpacaeval-hard | **91.6** | 90.0 |
| alpacaeval-length | 71.6 | **77.4** |
| donotanswer | **45.2** | 38.2 |
| hep-cpp | 78.7 | **82.6** |
| hep-go | 77.1 | **86.6** |
| hep-java | 73.5 | **82.9** |
| hep-js | 74.4 | **84.1** |
| hep-python | 79.0 | **85.7** |
| hep-rust | 74.4 | **81.4** |
| llmbar-adver-GPTInst | 23.4 | **24.5** |
| llmbar-adver-GPTOut | 63.8 | **67.0** |
| llmbar-adver-manual | **40.2** | 34.8 |
| llmbar-adver-neighbor | **20.9** | 16.8 |
| llmbar-natural | 66.0 | **74.5** |
| math-prm | 54.5 | **62.8** |
| mt-bench-easy | **92.9** | 87.5 |
| mt-bench-hard | **68.9** | 64.9 |
| mt-bench-med | **83.8** | 80.0 |
| refusals-dangerous | 71.5 | **74.0** |
| refusals-offensive | **76.0** | 73.5 |
| xstest-should-refuse | 70.5 | **71.8** |
| xstest-should-respond | 72.0 | **76.4** |

Table 8: Meta-Llama-3-70B-Instruct results on RewardBench

| Dataset | Vanilla | PINE |
|---|---|---|
| alpacaeval-easy | **100.0** | **100.0** |
| alpacaeval-hard | **100.0** | **100.0** |
| alpacaeval-length | **91.1** | 89.5 |
| donotanswer | 47.1 | **48.2** |
| hep-cpp | **92.7** | 92.1 |
| hep-go | 89.9 | **97.0** |
| hep-java | 92.1 | **97.0** |
| hep-js | 93.3 | **95.1** |
| hep-python | 90.9 | **95.4** |
| hep-rust | 89.0 | **91.5** |
| llmbar-adver-GPTInst | 55.4 | **57.6** |
| llmbar-adver-GPTOut | 73.4 | **76.6** |
| llmbar-adver-manual | **53.3** | 47.8 |
| llmbar-adver-neighbor | **32.8** | 28.7 |
| llmbar-natural | 83.0 | **84.0** |
| math-prm | 66.4 | **80.5** |
| mt-bench-easy | **100.0** | **100.0** |
| mt-bench-hard | **78.4** | 75.7 |
| mt-bench-med | **97.5** | **97.5** |
| refusals-dangerous | **63.5** | 62.5 |
| refusals-offensive | **66.5** | **66.5** |
| xstest-should-refuse | **68.8** | 63.3 |
| xstest-should-respond | **96.8** | 96.4 |

Table 9: Qwen1.5-1.8B-Chat results on RewardBench

| Dataset | Vanilla | PINE |
|---|---|---|
| alpacaeval-easy | **47.5** | 17.0 |
| alpacaeval-hard | **56.3** | 13.7 |
| alpacaeval-length | **50.0** | 45.8 |
| donotanswer | **54.0** | 52.6 |
| hep-cpp | 49.4 | **51.5** |
| hep-go | **54.3** | 48.8 |
| hep-java | **52.7** | 49.4 |
| hep-js | **48.2** | 47.6 |
| hep-python | 49.4 | **52.1** |
| hep-rust | **54.6** | 50.3 |
| llmbar-adver-GPTInst | 44.0 | **76.6** |
| llmbar-adver-GPTOut | **55.3** | 40.4 |
| llmbar-adver-manual | 44.6 | **64.1** |
| llmbar-adver-neighbor | 56.7 | **66.8** |
| llmbar-natural | 49.0 | **51.5** |
| math-prm | 45.5 | **70.2** |
| mt-bench-easy | 39.3 | **57.1** |
| mt-bench-hard | **54.1** | 35.1 |
| mt-bench-med | **46.2** | 45.0 |
| refusals-dangerous | 48.5 | **88.0** |
| refusals-offensive | 49.5 | **54.5** |
| xstest-should-refuse | 53.2 | **53.9** |
| xstest-should-respond | 52.2 | **68.8** |

Table 10: Qwen1.5-4B-Chat results on RewardBench

| Dataset | Vanilla | PINE |
|---|---|---|
| alpacaeval-easy | 59.5 | **77.5** |
| alpacaeval-hard | 62.1 | **80.5** |
| alpacaeval-length | 60.5 | **70.0** |
| donotanswer | **54.4** | 18.4 |
| hep-cpp | **50.0** | **50.0** |
| hep-go | 50.3 | **51.8** |
| hep-java | 49.1 | **51.2** |
| hep-js | **49.7** | 49.4 |
| hep-python | 50.0 | **53.0** |
| hep-rust | **50.6** | **50.6** |
| llmbar-adver-GPTInst | 36.4 | **38.6** |
| llmbar-adver-GPTOut | **54.3** | 51.1 |
| llmbar-adver-manual | **51.1** | 39.1 |
| llmbar-adver-neighbor | **52.2** | 42.5 |
| llmbar-natural | 48.0 | **53.5** |
| math-prm | 58.2 | **70.9** |
| mt-bench-easy | 44.6 | **75.0** |
| mt-bench-hard | **58.1** | 48.6 |
| mt-bench-med | **56.2** | 50.0 |
| refusals-dangerous | **43.0** | 31.0 |
| refusals-offensive | 47.0 | **71.0** |
| xstest-should-refuse | 53.6 | **64.3** |
| xstest-should-respond | 51.0 | **71.0** |

Table 11: Qwen1.5-7B-Chat results on RewardBench

| Dataset | Vanilla | PINE |
|---|---|---|
| alpacaeval-easy | 74.0 | **91.0** |
| alpacaeval-hard | 90.0 | **96.8** |
| alpacaeval-length | 65.8 | **74.7** |
| donotanswer | **19.9** | 11.0 |
| hep-cpp | 60.4 | **76.8** |
| hep-go | 61.9 | **69.2** |
| hep-java | 56.1 | **74.1** |
| hep-js | 59.5 | **68.6** |
| hep-python | 63.4 | **66.8** |
| hep-rust | 62.8 | **65.9** |
| llmbar-adver-GPTInst | **40.2** | 23.4 |
| llmbar-adver-GPTOut | **53.2** | 45.7 |
| llmbar-adver-manual | **35.9** | **35.9** |
| llmbar-adver-neighbor | **21.6** | 19.8 |
| llmbar-natural | **71.0** | 70.0 |
| math-prm | **57.9** | 55.8 |
| mt-bench-easy | **87.5** | 85.7 |
| mt-bench-hard | **62.2** | 55.4 |
| mt-bench-med | **77.5** | 72.5 |
| refusals-dangerous | **49.0** | 40.5 |
| refusals-offensive | **86.0** | **86.0** |
| xstest-should-refuse | **74.0** | 63.6 |
| xstest-should-respond | 75.6 | **86.6** |

Table 12: Qwen1.5-32B-Chat results on RewardBench

| Dataset | Vanilla | PINE |
|---|---|---|
| alpacaeval-easy | **97.0** | **97.0** |
| alpacaeval-hard | **98.9** | **98.9** |
| alpacaeval-length | 81.1 | **82.1** |
| donotanswer | **44.5** | 41.2 |
| hep-cpp | 87.8 | **91.5** |
| hep-go | 80.5 | **93.9** |
| hep-java | 88.7 | **96.3** |
| hep-js | 84.5 | **95.7** |
| hep-python | 86.3 | **93.6** |
| hep-rust | 82.0 | **88.1** |
| llmbar-adver-GPTInst | **43.5** | 34.8 |
| llmbar-adver-GPTOut | **68.1** | 57.4 |
| llmbar-adver-manual | 32.6 | **37.0** |
| llmbar-adver-neighbor | 25.0 | **28.4** |
| llmbar-natural | 83.0 | **85.0** |
| math-prm | 48.7 | **60.3** |
| mt-bench-easy | **96.4** | 92.9 |
| mt-bench-hard | **81.1** | 75.7 |
| mt-bench-med | 92.5 | **95.0** |
| refusals-dangerous | **80.0** | **80.0** |
| refusals-offensive | **99.0** | **99.0** |
| xstest-should-refuse | **90.6** | 89.3 |
| xstest-should-respond | 84.4 | **85.6** |

Table 13: Qwen1.5-72B-Chat results on RewardBench

| Dataset | Vanilla | PINE |
|---|---|---|
| alpacaeval-easy | **98.0** | **98.0** |
| alpacaeval-hard | 97.4 | **97.9** |
| alpacaeval-length | **85.3** | 84.2 |
| donotanswer | **39.0** | 38.2 |
| hep-cpp | 88.1 | **89.6** |
| hep-go | 85.4 | **92.1** |
| hep-java | 87.2 | **90.9** |
| hep-js | **90.9** | **90.9** |
| hep-python | 87.2 | **89.6** |
| hep-rust | **88.1** | 87.2 |
| llmbar-adver-GPTInst | **44.6** | 33.7 |
| llmbar-adver-GPTOut | 57.4 | **61.7** |
| llmbar-adver-manual | **41.3** | 39.1 |
| llmbar-adver-neighbor | **28.0** | 20.9 |
| llmbar-natural | **84.0** | 81.0 |
| math-prm | **48.5** | 47.9 |
| mt-bench-easy | 96.4 | **100.0** |
| mt-bench-hard | **70.3** | 62.2 |
| mt-bench-med | **95.0** | 92.5 |
| refusals-dangerous | **75.5** | 73.0 |
| refusals-offensive | 94.0 | **95.0** |
| xstest-should-refuse | 87.7 | **91.2** |
| xstest-should-respond | **86.8** | 85.0 |

Table 14: Qwen2.5-72B-Instruct results on RewardBench

| Dataset | Vanilla | PINE |
|---|---|---|
| alpacaeval-easy | **99.0** | **99.0** |
| alpacaeval-hard | 97.9 | **98.9** |
| alpacaeval-length | **91.6** | 89.5 |
| donotanswer | 48.5 | **52.9** |
| hep-cpp | **95.7** | **95.7** |
| hep-go | 97.0 | **98.8** |
| hep-java | **98.8** | 97.6 |
| hep-js | 94.5 | **98.2** |
| hep-python | **98.8** | **98.8** |
| hep-rust | 94.5 | **97.6** |
| llmbar-adver-GPTInst | 66.3 | **68.5** |
| llmbar-adver-GPTOut | **76.6** | 72.3 |
| llmbar-adver-manual | **65.2** | 63.0 |
| llmbar-adver-neighbor | 41.8 | **44.0** |
| llmbar-natural | **92.0** | 87.0 |
| math-prm | 74.5 | **84.8** |
| mt-bench-easy | **100.0** | **100.0** |
| mt-bench-hard | **94.6** | 91.9 |
| mt-bench-med | 97.5 | **100.0** |
| refusals-dangerous | 78.0 | **82.0** |
| refusals-offensive | **95.0** | 92.0 |
| xstest-should-refuse | **92.2** | 89.6 |
| xstest-should-respond | **95.2** | 93.6 |

Table 15: Qwen1.5-110B-Chat results on RewardBench

| Dataset | Vanilla | PINE |
|---|---|---|
| alpacaeval-easy | 95.0 | **97.0** |
| alpacaeval-hard | **98.9** | **98.9** |
| alpacaeval-length | **93.7** | 88.4 |
| donotanswer | 51.5 | **55.9** |
| hep-cpp | 87.8 | **92.1** |
| hep-go | 83.8 | **94.8** |
| hep-java | 86.6 | **94.8** |
| hep-js | 90.5 | **92.4** |
| hep-python | 83.8 | **93.9** |
| hep-rust | 85.7 | **90.9** |
| llmbar-adver-GPTInst | **70.1** | 65.2 |
| llmbar-adver-GPTOut | **72.3** | 61.7 |
| llmbar-adver-manual | 60.9 | **65.2** |
| llmbar-adver-neighbor | **44.8** | 41.4 |
| llmbar-natural | 86.5 | **90.0** |
| math-prm | 69.6 | **79.2** |
| mt-bench-easy | 98.2 | **100.0** |
| mt-bench-hard | **83.8** | **83.8** |
| mt-bench-med | **97.5** | **97.5** |
| refusals-dangerous | 76.0 | **84.0** |
| refusals-offensive | **97.0** | **97.0** |
| xstest-should-refuse | 91.6 | **91.9** |
| xstest-should-respond | **95.6** | 92.4 |

# E    IMPLEMENTATION DETAILS

## E.1    EXPERIMENT SETTING

For reproducibility, the generation temperature is set to 0. We use PyTorch (Ansel et al., 2024; Paszke et al., 2019),[7] Transformers (Wolf et al., 2020),[8] and vLLM (Kwon et al., 2023) for our experiments.[9] All experiments are launched with a single node of 8x A100 80G with SXM connection. 70B and 110B models are launched with 3x and 4x A100, and other model sizes can be launched with 1x A100.

## E.2    MOLECULE GENERATION AND MATH REASONING TASK DETAILS

**Molecule Generation.** In this task, the input contains several properties that are interchangeable, and LMs are asked to generate molecules that satisfy these properties. We train such an LM with QM9 (Ramakrishnan et al., 2014) dataset. The QM9 dataset collects over $130k$ 3D molecules with 3D structures (Li et al., 2024) calculated by density functional theory (DFT). Each molecule in QM9 has less than 9 heavy atoms, and its chemical elements all belong to H, C, N, O, F. We take six quantum property values as the conditional input to LMs and train LMs to generate molecules with the conditioned quantum property values. We split the training dataset of QM9 to two subsets where each subset has $50k$ samples, and train LMs and an EGNN-based quantum property prediction models (Satorras et al., 2021) on these two subsets, respectively. The six quantum properties are polarizability ($\alpha$), HOMO energy ($\epsilon_{\text{HOMO}}$), LUMO energy ($\epsilon_{\text{LUMO}}$), HOMO-LUMO gap ($\Delta\epsilon$), dipole moment ($\mu$) and heat capacity at 298.15K ($C_v$). The LM is a 8-layer Llama model with 8 attention heads and 768 hidden dimensions. To evaluate the performance, we sample 10000 sets of 6-property conditions, randomize the property order in each condition, and generate molecules conditioned on these property values by the trained LM, and compute the mean absolute difference (MAE) between the given property values and the property values of the generated molecules. Note that we use the trained EGNN-based property prediction models to calculate the property values of the generated molecules.

**Math Reasoning.** We use R-GSM (Chen et al., 2024a), a subset of GSM8K. This small dataset (which contains 220 problems) is designed to test LMs' performance with interchangeable premise orders. Problems in the dataset contain several conditions that do not have a progressive relationship. Therefore, their positions are interchangeable. We further clean this dataset to remove problems where conditions do not read smoothly after changing positions (e.g., use pronouns in the first condition but introduce names in the second condition), yielding a small set containing 95 problems. We test Qwen-1.5 models on this dataset.

## E.3    PROMPTS

We use the prompts provided by RewardBench (Lambert et al., 2024a) official repo for the LM-as-a-judge task:

**System prompt**:

```
Please act as an impartial judge and evaluate the quality of the
responses provided by two AI assistants to the user question
displayed below.  You should choose the assistant that follows
the user's instructions and answers the user's question better.
Your evaluation should consider factors such as the helpfulness,
relevance, accuracy, depth, creativity, and level of detail of
their responses.  Begin your evaluation by comparing the two
responses and provide a short explanation.  Avoid any position
biases and ensure that the order in which the responses were
presented does not influence your decision.  Do not allow the
length of the responses to influence your evaluation.  Do not
```

---

[7]Customized license. https://github.com/pytorch/pytorch
[8]Apache-2.0 license. https://huggingface.co/docs/transformers/en/index
[9]Apache-2.0 license. https://github.com/vllm-project/vllm.

```
favor certain names of the assistants.  Be as objective as
possible.  After providing your explanation, output your final
verdict by strictly following this format:  "[[A]]" if assistant A
is better, "[[B]]" if assistant B is better.
```

**User prompt**:

```
[User Question]

......

[The Start of Assistant A's Answer]

......

[The End of Assistant A's Answer]

[The Start of Assistant B's Answer]

......

[The End of Assistant B's Answer]
```

We use the prompts of official repo (Liu et al., 2024) for the retrieval-augmented QA experiments:

**User Prompt**:

```
Write a high-quality one-sentence answer for the given question
using only the provided search results (some of which might be
irrelevant).

Document (Title:  ......):  ......

Document (Title:  ......):  ......

......

Document (Title:  ......):  ......

Document (Title:  ......):  ......

Question:  ......
```

Molecule generation does not need prompts. We use prompts in OpenAI/simple-evals [10] to evaluate R-GSM dataset.

# F  QUALITATIVE EXAMPLES

## F.1  LM-AS-A-JUDGE

```
[User Question]

Who created the Superman cartoon character?

[The Start of Assistant A's Answer]

The Superman cartoon character was created by Jerry Siegel and Joe
Shuster in 1938.

[The End of Assistant A's Answer]

[The Start of Assistant B's Answer]

Superman, the iconic comic book superhero, was created by writer
Jerry Siegel and artist Joe Shuster.  Superman first appeared
in Action Comics #1, which was published by Detective Comics,
Inc.  (later DC Comics) in June 1938.  The character's immense
```

---

[10]https://github.com/openai/simple-evals

popularity established him as one of the most enduring and recognizable figures in the superhero genre.

[The End of Assistant B's Answer]

Here are outputs of Qwen1.5-7B-Chat:

**Vanilla**:       Both AI assistants provide accurate and informative responses to the user's question.  Assistant A's answer is brief and directly states the creators' names and the year of creation, while Assistant B offers additional context by mentioning the first appearance, the publisher, and the character's significance in the superhero genre.  However, since the user asked for the creators, Assistant A's answer is more directly helpful in this specific context.  Final verdict:  [[A]]

**Vanilla   (Swapped)**:      [[B]]. Assistant B's response is more detailed and informative, providing additional context about the origin of the character in Action Comics #1, the publisher, and its significance in the superhero genre.  It also includes the names of the creators, which is crucial information for the question.  Assistant A's response is shorter and lacks the historical context and specific details.  Therefore, Assistant B is better in answering the user's question.

**PINE**:           [[B]]. Assistant B provides a more comprehensive and detailed response to the user's question.  It not only mentions the creators but also provides additional information about the character's debut and the company that published the comic.  This answer is not only accurate but also offers context that highlights the significance of the creation.  Assistant A's response is brief and lacks the depth and historical context provided by Assistant B.

## F.2   RETRIEVAL-AUGMENTED QUESTION-ANSWERING

**User Prompt**:

Document (Title:  Pearl) south Pacific and Indian Ocean.  The largest pearl oyster is the "Pinctada maxima", which is roughly the size of a dinner plate.  South Sea pearls are characterized by their large size and warm luster.  Sizes up to 14 mm in diameter are not uncommon.  In 2013, Indonesia Pearl supplied 43 percent of South Sea Pearls international market.  The other significant producers are Australia, Philippines, Myanmar and Malaysia.  In 1914, pearl farmers began growing cultured freshwater pearls using the pearl mussels native to Lake Biwa.  This lake, the largest and most ancient in Japan, lies near the city of Kyoto.  The

Document (Title:  Laccadive Sea) the gulf as most productive in the world.  Although extraction of natural pearls is considered too expensive in most parts of the world, it is still conducted in the gulf.  Also collected in large numbers are Shankha mollusks ("Xancus pyrum") whose shells are used as a ritual and religious object.  Other mollusks of the sea are either too scarce or not popular in the Indian society and therefore have no commercial value.  Another traditional occupation in the Laccadive Sea is fishing.  The annual fish catch is 2,000 to 5,000 tonnes from the Lakshadweep islands, which is mostly constituted by tuna

Document (Title:  Pearl) including the Cook Islands and Fiji are being extensively used for producing cultured pearls.  The rarity

of the black cultured pearl is now a "comparative" issue.  The black cultured pearl is rare when compared to Chinese freshwater cultured pearls, and Japanese and Chinese akoya cultured pearls, and is more valuable than these pearls.  However, it is more abundant than the South Sea pearl, which is more valuable than the black cultured pearl.  This is simply because the black pearl oyster "Pinctada margaritifera" is far more abundant than the elusive, rare, and larger south sea pearl oyster "Pinctada maxima", which cannot

Document (Title:  Pearl powder) Pearl powder Pearl powder () is a preparation of crushed pearls used in China and elsewhere for skin care and in traditional Chinese medicine.  Pearl powder is made from freshwater pearls or saltwater pearls below jewellery grade.  These are sterilised in boiling water and then milled into a fine powder using stainless steel grinding discs or by milling with small porcelain balls in moist conditions.  The powder is sold as such or mixed into creams.  Pearl powder is widely believed to help improve the appearance of the skin, and is used as a cosmetic by royal families in Asia.  It

Document (Title:  Hyderabad pearl) with white pearls.  Recently, several pearl makers are exporting processed pearls to markets in Europe and the US. With the capital that they gain from this marketing, they are able to purchase machinery for advanced refinement.  In particular, equipment that uses enzymes present in thermophiles is able to substantially improve the process of refining pearls.  Hyderabad pearl Hyderabad is considered the main pearl trading center in India.  The most notable area devoted to the trade is the village called Chandanpet just outside Hyderabad, wherein almost the entire population is engaged in the delicate art of drilling pearls, a skill they

Document (Title:  Pearl) pearls".  The correct definition of a South Sea pearl { as described by CIBJO and GIA { is a pearl produced by the "Pinctada maxima" pearl oyster.  South Sea pearls are the color of their host "Pinctada maxima" oyster { and can be white, silver, pink, gold, cream, and any combination of these basic colors, including overtones of the various colors of the rainbow displayed in the pearl nacre of the oyster shell itself.  South Sea pearls are the largest and rarest of the cultured pearls { making them the most valuable.  Prized for their exquisitely beautiful órientór lustre,

Document (Title:  Chandrani Pearls) year 2007{08 Chandrani Pearls imported their pearls from Japan, China or Korea.  Chandrani Pearls Chandrani Pearls is a prominent pearl jewelery brand of India.  It pioneered the concept of pearls in India.  Chandrani Pearls's headquarters is at Kolkata in West Bengal.  Chandrani Pearls was started on 24 January 1985 by Mr.  Kuldip Nayar, his wife Mrs.  Lakshmi Nayar and his father late Mr.  N.C. Nayar in Kolkata's up market Minto Park area.  Chandrani Pearls management is now assisted by Nisheeth Nayar, sons of Mr.  Kuldip Nayar.  Chandrani Pearls have 63 showrooms across 9 states.  From a modest turnover of Rs.

Document (Title:  Pearl) For thousands of years, seawater pearls were retrieved by divers in the Indian Ocean in areas such as the Persian Gulf, the Red Sea and the Gulf of Mannar.  Evidence also suggest a prehistoric origin to pearl diving in these regions.  Starting in the Han Dynasty (206 BC{220 AD), the Chinese hunted

extensively for seawater pearls in the South China Sea. In the
14th-century Arabian Sea, the traveller Ibn Battuta provided the
earliest known description of pearl diving by means of attaching a
cord to the diver's waist.

Document (Title: Chandrani Pearls) Chandrani Pearls Chandrani
Pearls is a prominent pearl jewelery brand of India. It pioneered
the concept of pearls in India. Chandrani Pearls's headquarters
is at Kolkata in West Bengal. Chandrani Pearls was started on
24 January 1985 by Mr. Kuldip Nayar, his wife Mrs. Lakshmi
Nayar and his father late Mr. N.C. Nayar in Kolkata's up market
Minto Park area. Chandrani Pearls management is now assisted by
Nisheeth Nayar, sons of Mr. Kuldip Nayar. Chandrani Pearls have
63 showrooms across 9 states. From a modest turnover of Rs. 2.75
lakhs in 1990 to a turnover of Rs. 15.00 crores in

Document (Title: Hyderabad pearl) Hyderabad pearl Hyderabad
is considered the main pearl trading center in India. The most
notable area devoted to the trade is the village called Chandanpet
just outside Hyderabad, wherein almost the entire population is
engaged in the delicate art of drilling pearls, a skill they have
practiced for generations. This practice also makes the city
one of the largest pearl drilling locations in India. The pearl
industry in Hyderabad flourished due to the patronage of the Qutub
Shahi kings and the Asaf Jahis, who were said to have an affinity
for sparkling jewels. The pearls were not only part of

Question: in which sea pearl is found in india?

Here are Llama-3-70B-Instruct results:

**Vanilla**: According to the search results, pearls are found in the
Laccadive Sea in India.

**PINE**: Pearls are found in the Indian Ocean, specifically in the
Gulf of Mannar and the Laccadive Sea, off the coast of India.

## F.3 MOLECULE GENERATION

The task has properties as input, and molecule structure as output. Below is one example:

**Input**: cv*-30.887* alpha*81.68* mu*2.0332* homo*-5.8749* gap*7.8668*
lumo*1.9919*

**Output**: H 0.00 0.00° 0.00° O 0.96 1.57° 0.00° C 1.95 2.33° 0.00° H
2.35 2.50° 0.70° H 2.32 2.53° -0.71° C 3.22 2.00° -0.02° H 3.46
1.80° 0.24° C 4.50 2.23° 0.00° H 4.99 2.22° 0.26° H 4.55 2.46°
-0.12° C 5.58 2.08° -0.18° H 6.55 2.07° -0.09° N 5.89 2.05° -0.46°
C 5.57 2.18° -0.71° H 4.57 2.25° -0.78° H 6.29 2.32° -0.72° H 5.98
2.06° -0.86° C 5.26 1.85° -0.34° H 6.06 1.72° -0.33° C 3.81 1.74°
-0.36° H 3.94 1.47° -0.29° H 3.58 1.75° -0.65°

## F.4 R-GSM

R-GSM is just a subset of GSM8K, with the premise order changes. Here is an example input:

**Input**: Carmen goes to an auction to win an antique desk. The bids
on the desk rise by 50 each time and 3 other people each bid once.
She accepts the opening bid of 200 and continues bidding until she
wins. Carmen bids after each of the 3 other people and eventually
wins. How much money, in dollars, does the desk cost her?

Here    The bids on the desk rise by 50 each time and 3 other people each bid once. and She accepts the opening bid of 200 and continues bidding until she wins. are interchangeable.

