# OpenReview forum: "Eliminating Position Bias of Language Models: A Mechanistic Approach"
_ICLR.cc/2025/Conference — ICLR 2025 Poster_

### Official Review · Reviewer_1nRN · 2024-11-04

**Soundness:** 3
**Presentation:** 3
**Contribution:** 3
**Rating:** 6
**Confidence:** 3

**Summary:**

This work addresses the position bias problem in language models (LMs), where models prioritize information based on its order in the input, affecting performance and reliability across applications. Through a mechanistic analysis, the study identifies causal attention and relative positional encodings as primary contributors to this bias. To mitigate it, the authors introduce Position-INvariant InferencE (PINE), a zero-shot approach that replaces causal attention with bidirectional attention between documents and uses attention values to determine document order. PINE effectively enhances model performance in tasks like QA, molecule generation, and math reasoning, showing notable gains in reasoning benchmarks where it surpasses even state-of-the-art models like GPT-4 in specific evaluations.

**Strengths:**

1. This paper proposes a simple yet effective and novel method to mitigate the postion bias problem in LLMs.

2. Extensive experiemnts on Llama-3 and Qwen demostrate the effectiveness of the proposed method.

**Weaknesses:**

1. The ablation study of the strategy of re-assign positions is required to better discuss the potential of further improving the method.

2. There is no baseline comparison with other calibration based methods.

**Questions:**

1. Do different position assigning strategies have a big effect on the downstream performance?

---

> ### Author Response · Authors · 2024-11-22
> **Rebuttal (1/1)**
>
> We thank you for your advice! We are happy that you find our method simple, effective and useful. We address your concerns below:
>
> > W1 & Q1: The ablation study of position re-assignment
>
> In Figure 4, we compare our re-assignment method with the reversed one,  random shuffle one, and positions unchanged one. Results show that our methods achieve the best.
>
> Since testing all available re-assignment methods is impossible, we exclude alternatives that lack supporting evidence (for example, re-assigning by counting overlapping of bigram with question), or need extra efforts (for example, training an NN to do re-assignment). Therefore, Figure 4b contains all plausible methods that we can come up with. We are willing to add more experiments if you have any suggestions.
>
> At last, we hope to point out that our re-assignment method aligns with research on attention interpretability work such as retrieval heads [1].
>
> [1] Retrieval Head Mechanistically Explains Long-Context Factuality
>
>
> > W2: Comparison with calibration-based methods
>
> We thank for pointing out this discussion. First, we compare a calibration method [2] in L459 and find it yields non-sense results due to its strong hypothesis. Another calibration method that is relevant to our method is [3]. However, we are unable to deliver its results since it does not publicly release the codes. We also face difficulties implementing the method ourselves since papers cannot cover every detail: specifically, the paper mentions how to manipulate attention when decoding tokens but does not mention details about encoding documents themselves (i.e., the first forward pass when decoding).
>
>
>
> [2] Calibrate Before Use: Improving Few-shot Performance of Language Models
>
> [3] Found in the Middle: Calibrating Positional Attention Bias Improves Long Context Utilization
>
>
> We hope our explanation addresses your concerns, and we are happy to discuss them with you if you have any follow-up questions.

---

### Official Review · Reviewer_1Z1E · 2024-11-05

**Soundness:** 4
**Presentation:** 4
**Contribution:** 4
**Rating:** 8
**Confidence:** 3

**Summary:**

Proposed work introduces PINE, an attention mechanism that computes the attention scores in a positionally invariant manner. The method does not require any training and can be plugged into an transformer inference setup with minimal costs. Improved results in multiple benchmarks show the effectiveness of the approach.

**Strengths:**

- The approach does not require any extra training, and can be used by any attention mechanism.
- Remarkable improvements on rewardbench evals.
- Comparison with relevant baselines and experiments with multiple models performed.
- Positional invariancy is thoroughly ascertained through empirical and theoretical explanations.

**Weaknesses:**

- Regarding,
> Hsieh et al. (2024) assumes that the position bias and real relevance are linear combinations and propose solutions accordingly. Different from them, we aim to eliminate the position bias from the mechanical perspective without any assumption at a reasonable cost.

Although it is claimed that this solution is better motivated than the cited one, a comparison is still required since both papers are solving the same problem and show very good improvements in similar yet different tasks.


- [This](https://openreview.net/pdf?id=gEMLMMG0m9) work debiases positional bias in the attention matrix by averaging attention values of documents at different positions. A mention of why such simple methods to debias are less relevant or a comparison is required.


- Regarding line 288,
> the extra big O computation complexity to obtain hidden states is O(nk log k),

From my understanding, for an end-to-end inference time overhead calculation, this should be multiplied by #layers in the model. And further with #heads if computation across heads is not parallelized. The equation does not capture this.
Hence for lines 518-519,
> we find the wall time of PINE is ~2x and ~8x of the vanilla inference

These numbers should depend on the size of the models being considered. Can you tabulate this for each model separately to give a better picture of overhead time? This is significant as benchmarks tend to use large models as evaluators and an inference time $\propto $ #layers limits the practical applicability of the work.

**Questions:**

<none>

---

> ### Author Response · Authors · 2024-11-22
> **Rebuttal (1/1)**
>
> We thank you for your helpful advice! We sincerely appreciate your favor to our paper and are happy to see you find our paper general and useful.
>
> > W1: Comparison with Hsieh et al.
>
> This is indeed a relevant method. However, we are unable to deliver results since it does not publicly release codes yet. We also face difficulties implementing the method since paper cannot cover every detail due to length limits.
>
> For example, the paper mentions how to manipulate attention when decoding tokens but does not mention details about encoding documents themselves (i.e., the first forward pass when decoding).
>
>
> > W2: Averaging importance score
>
> We briefly talked about using averaging over summation in our paper (L231) to prevent putting higher scores on longer documents. In our pilot experiments, we find that summation converts models from position bias to length bias.
>
> We also try maximum instead of averaging and find this method usually has noticeably worse performance than averaging possibly due to noises brought by unimportant tokens. Therefore, we chose averaging in our final version.
>
> Thanks for pointing out this discussion. We’ve added this discussion to the paper's Section 3.3 footnote, which is highlighted in green.
>
> > W3: Time cost w.r.t. Layers
>
> We want to point out a slight mistake: The time gain ratio compared with vanilla inference is independent w.r.t. Layers. For example, if each layer of PINE needs 2x time, then in total, PINE still needs 2x time regardless of layers because PINE and vanilla inference have the same number of layers.
>
> However, we agree that the architecture parameters may affect the ratio factor, such as the compute balance between attention and FFN.
>
> In our experiments, we find that if the number of input documents is 2, and time is ~2x across sizes consistently, and if the number of input documents is 20, the time is ~8x across sizes consistently. We can only get a rough ratio since the real ratio depends on factors such as GPU types, connection type, model sharding type, batch size, server burden, etc.
>
>
> We hope our response addresses your concerns and are happy to discuss if you have any follow-up questions!

---

> > ### Comment · Reviewer_1Z1E · 2024-11-28
> >
> > > implementing the method since paper cannot cover
> >
> > Although the code is not public, the evaluation datasets and code is. Any comparison on the same benchmark should be enough to confirm the gains.
> >
> > > We briefly talked about using averaging over summation in our paper (L231) to prevent putting higher scores on longer documents.
> >
> > I understand the need of averaging instead of summation. What I wanted to express is that, the cited work reorders the input several times and computes the score each time for the same document to account for the position bias. This is a simple way to avoid it. I wanted to understand if such a method is enough. The proposed method and this "naive" method, both require more computation than vanilla attention. If we match the number of operations, do you think your method is "better"?  Just wanted a comment on that.
> >
> > > We want to point out a slight mistake:
> >
> > My apologies, that was a poorly thought out comment.

---

> > > ### Author Response · Authors · 2024-12-02
> > >
> > > Thanks for your additional valuable feedback!
> > >
> > > > Implementation of Hsieh et al.
> > >
> > > Although the evaluation code is publicly available, we tried and could not reproduce the paper-reported number perfectly on the vanilla inference model, which is probably due to the slight difference in the prompts, etc.
> > >
> > > Therefore, we implemented the method ourselves and ran it on the same prompts we used in our experiments (although we admit we can not guarantee the reproduction is perfectly correct since no public codes and some missing details in the paper). Our results show that our approach performs better than Hsieh's, probably because the strong assumption in Hsieh's does not hold well in every case.
> > >
> > > On Llama 3 8B Instruct and RewardBench:
> > >
> > > | Method |  Accuracy |
> > > | - | - |
> > > | Vanilla (Shuffle) | 64.8|
> > > | Peysakhovich & Lerer (2023) [k=1] | 65.2 |
> > > | Hsieh et al. | 65.0 |
> > > | PINE | **66.7** |
> > >
> > > We hope this result can address your concerns.
> > >
> > > > If we match the number of operations, do you think your method is "better"?
> > >
> > > The method you mention is Peysakhovich & Lerer (2023). We implemented the method by ourselves (since the code is not publicly available) and reported the number in the above table. We find the method cannot beat PINE.
> > >
> > > However, we have to admit that the upper bound of such prompt order optimization is higher than PINE since the best case is that the ground-truth documents always appear at the correct position. We believe PINE is still a better choice until such optimization method is discovered, as the above results suggest.
> > >
> > > We hope our additional experiments can answer your questions!

---

### Official Review · Reviewer_Qpxn · 2024-11-05

**Soundness:** 3
**Presentation:** 3
**Contribution:** 3
**Rating:** 6
**Confidence:** 3

**Summary:**

The paper presents a modification to the causal attention mechanism of decoder-based transformers. The proposed mechanism adds bidirectional attention to each document, in order to make representations position-invariant. New bidirectional attention terms are added in blocks (for the tokens corresponding to each document), which are ordered according to their importance to the document on the diagonal. The proposed importance score is based on a version of the standard attention calculation that ignores positional differences. The paper proves positional invariance and evaluates the empirical performance of the new method on a set of tasks.

**Strengths:**

1. The paper is well-written and the graphics are appealing.
2. Position-invariant representations are nice, and you obtain them without k! overhead.
3. The positional bias on RewardBench appears significant and adds credibility to your argument that position bias is an important problem.
4. Strong results for the Qwen models on RewardBench.
5. Strong results for Llama on the reasoning subset of RewardBench.

**Weaknesses:**

1. You compare your method to other mechanistic methods with different goals: for example, PCW helps extend the context window; NIA gives a speed boost by making the attention computation sub-quadratic. Thus, these are both *approximate* methods. Your method, by contrast, introduces extra overhead compared to standard sampling. For a fair comparison, it is important to look for methods that are more resource-intensive than standard sampling (for example, if your method has 8x overhead, a simple baseline could involve sampling 8 permutations of the documents and running vanilla sampling 8 times, followed by some aggregation scheme like majority voting).

2. I appreciate the analysis shown in Figure 4b) but I would also like to see an analysis of the choice of importance score (see question 1. below).

3. On both RAG-QA and RewardBench, PINE does not seem to improve over vanilla sampling with the Llama models. To see this, note that the average of the GT-A and GT-B scores should give the expected performance for vanilla sampling (with randomized ordering of the two documents -- please correct me if I am wrong). On molecular generation and math reasoning, you switch between evaluating the Llama and Qwen models, which does not inspire confidence; I would like to see the same list of models evaluated for each task. Overall, your selection of tasks seems somewhat contrived. See question 2. below.

**Questions:**

1. Your method gives position invariance as long as the attention orderings for each document depend only on the content of the documents rather than their positions. Thus, mathematically you can consider any function that maps a list of k documents to a permutation on k-1 documents (k-1 since the document on the diagonal is always in the last position). This class of functions is large and includes special cases like variants of your importance score (e.g., variants regarding the choice of normalization). It seems important to assess the sensitivity of PINE with respect to the choice of ordering function.

2. If position bias is as big a problem for our field as the paper argues, it should be possible to show gains on mainstream benchmarks. Consider the generic position-dependent biases in few-shot prompting, as identified by Zhao et al. in their contextual calibration paper. Could you improve, say, the recency bias of few-shot prompting with your approach? If yes, it would be nice to see this demonstrated on a mainstream LLM benchmark.

3. I am not very familiar with the Qwen series of models. Since your method yields essentially no gains for Llama on RewardBench (as noted above – please correct me if my analysis is mistaken), I wonder if the strong performance on the Qwen models has something to do with the architecture of these models. After looking into the matter, I wonder if the Qwen models’ use of sliding window attention could be the reason why the extra bi-directional attention from your method yields a benefit. What do you think?

4. The ability to permute documents gives you a way to estimate the variance of vanilla sampling performance. Given the wide disparity between the GT-A and GT-B performances on RewardBench, did you look into randomizing the order of premises on R-GSM? This seems especially important because there are so few samples (only 95) in this data set.

---

> ### Author Response · Authors · 2024-11-22
> **Rebuttal (1/1)**
>
> We thank you for your detailed and helpful suggestions and are glad you found our presentation clear and the results strong. We address your concerns below.
>
> > W1: Baseline comparison
>
> PCW and NIA were initially designed for long ICL tasks, but they can all mathematically guarantee the elimination of position bias (similar to our proof). Therefore, they are not approximations of elimination (hopefully, we correctly understand the meaning of approximation in your context; please correct us if it is wrong). We compare our methods with permutation and calibration baselines in the paragraph of L459. In short, the calibration-based method has non-sense outputs, which we believe is caused by its strong assumption. Permutation-based methods perform worse than ours. Note that the permutation baselines are only conducted in RewardBench since, in RAG, the computation cost of O(10!) and O(20!) is intractable. We apologize for not making this clearer, and we’ve highlighted permutation results in our updated version.
>
> > W2 & Q1: the different choice of importance score
>
> This is a very interesting question, and indeed, there are a lot of available mappings. In Figure 4, we compare our importance score with the reverse score, random scores, and positions unchanged. We use attention scores as importance scores, which aligns with research on attention interpretability work such as retrieval head [1]. Since it is not tractable to test all possible functions, we exclude alternatives that lack supporting evidence (for example, overlapping of bigram with question), or need extra efforts (for example, training an NN to do ranking). Therefore, we do not find other plausible importance scores to compare besides Figure 4b. We are willing to add more experiments if you have any suggestions.
>
> [1] Wu, Wenhao, et al. "Retrieval head mechanistically explains long-context factuality." arXiv preprint arXiv:2404.15574 (2024).
>
>
> > W3 & Q2: Performance
>
> * The shuffle results are average of GT at A and GT at B
>
> Vanilla (Shuffle) is not necessarily the average of two extreme cases. The key problem is that the model may prefer different positions for different problems. For example, if the dataset has ten questions and for the first five it prefers A and second five it prefers B, then GT at A and GT at B will all have 50% acc. However, Vanilla (shuffle) may have up to 100% and low to 0%, depending on the concrete shuffle. In Table 2, the shuffled version is the third line, and our methods show consistent improvements across Llama/Qwen and different sizes.
>
> * Switching models between models in molecule generation
>
> We initially hope to show that our experiments on diverse models and tasks work, and we apologize for causing such confusion. The llama and Qwen models have no differences in the molecule generation tasks because we train from **scratch**, and the two models have indistinguishable architecture differences (if not all the same). For the math dataset, the llama results are shown below, where PINE performs better than the baseline.
>
> |Method| 8B | 70B |
> |-|-| -|
> |Vanilla| 63.2 | 84.2 |
> |PINE|**64.2** |**86.3** |
>
> * Few-shot prompting experiments
>
> We conducted a few-shot experiment on the ARC dataset with the ICL example selection same as in previous work [2]. Experiments were conducted on the Llama 3 8B model, and PINE showed the best performance.
>
> |Method| Vanilla | Permutation (Cylic) | PINE |
> |-|-| -| -|
> |0-shot| 80.34| N/A | N/A |
> |3-shots|80.34 | 79.48 | **80.69**|
> |5-shots|79.91 | 80.34 | **80.52** |
>
>
>
> [2] https://arxiv.org/abs/2309.03882
>
> > Q3: Llama performance and Qwen architecture
>
> We hope our response to W1 clarifies your first subquestion about Llama's performance. In short, we need to compare the last two rows (i.e., PINE and Vanilla), and our methods have noticeable improvements in Llama. The Qwen architecture uses the same architecture as Llama, such as GQA, causal attention, RoPE, layernorm, and MLP. We do not use sliding window features, as all inputs/outputs do not exceed window length.
>
> > Q4: R-GSM variance
>
> We agree that R-GSM has a high variance due to the small data size, and we think this experiment is more like a bonus experiment. Since each problem in R-GSM may have a different number of conditions, and we do not have prior information about the best and worst order of conditions, we are not able to present numbers such as GT at A and GT at B.  If we random shuffle several times and take the average of vanilla inference performance, then we have the following results (PINE vs Avg of Vanilla) :  82.1 vs. 79.5 in Qwen 110B, and 50.5 vs. 46.8 in Qwen 7B.
>
> We conclude that our methods generally have better or on-par performance while maintaining 0 variances, whereas the vanilla inference encounters a high variance ( 6%~7% deviation in accuracy on average).
>
>
> We hope our responses address your concerns, and we are happy to discuss any follow-up questions.

---

> > ### Comment · Reviewer_Qpxn · 2024-11-26
> >
> > I appreciate the additional results addressing my previous questions. I'd like to discuss two points:
> >
> > **Regarding baselines**: when I describe PCW and NIA as "approximate," I'm referring to how they reduce computational requirements by simplifying the attention mechanism - specifically by eliminating inter-document attention. These methods prioritize computational efficiency to handle longer sequences. Your approach, however, maintains full quadratic attention plus additional computational costs, suggesting it's not designed for processing longer sequences. Given that you're testing on standard-length sequences where computational compromises aren't necessary, it would be more appropriate to compare against methods intended for standard lengths.
> >
> > About permutation baselines: the methods you cite (contextual calibration and PriDe) try to learn order bias in order to remove it at inference-time. I'm suggesting a simpler baseline:
> >
> > answers = [ ]
> >
> > Sample k permutations [σ₁, σ₂, ..., σₖ]
> >
> > **for** permutation in [σ₁, σ₂, ..., σₖ]:
> >
> > &nbsp;&nbsp;&nbsp;&nbsp;permuted_documents ← permute(documents, permutation)
> >
> > &nbsp;&nbsp;&nbsp;&nbsp;answer ← run_model(query, permuted_documents)
> >
> > &nbsp;&nbsp;&nbsp;&nbsp;answers.append(answer)
> >
> > **return** majority_vote(answers)
> >
> > This method requires no training and should not give "rubbish" outputs as you observe with the other permutation-based methods. It introduces a k-fold computational overhead, but helps reduce variance from positional bias. You can set k=8 to match your method's computational overhead. When majority voting isn't applicable, you can substitute another aggregation method or random selection of the answer.
> >
> > **Regarding shuffle performance**: I do not completely follow your explanation. I believe we agree that vanilla shuffle's expected value is (1/2)*E[GT-A] + (1/2)*E[GT-B]. Consider randomly sampling 1,000 examples from the data-generating distribution - approximately 500 will be assigned to GT-A and 500 to GT-B. The expected values for GT-A and GT-B match what you'd get from independent 50% resamples of your original dataset, so E[shuffle] = (1/2)*E[GT-A] + (1/2)*E[GT-B]. Any deviation from this theoretical expectation indicates the impact of finite sample effects. Would you agree with this analysis?
> >
> > Since your experiments provide performance data for both GT-A and GT-B on each question, you can simulate vanilla shuffle computationally by randomly selecting either GT-A or GT-B results for each question. This eliminates the need for additional model runs - you can simply randomly select from your existing correctness data. For a single simulation, you flip a coin for each example to choose GT-A or GT-B. Running this simulation, say, 10,000 times should provide good estimates of the mean and variance of vanilla shuffle. Could you report the performance of vanilla shuffle as mean +/- standard deviation, rather than as a single number?

---

> ### Author Response · Authors · 2024-11-27
>
> Thanks for reading our rebuttal. We are happy that we addressed your questions, and here are our responses to your new feedback:
>
> > The permutation baseline
>
> We apologize for not making the permutation baseline clear. In the L459, we report two methods: calibration methods does not work, and permutation (which is exactly what you describe) methods that underperform ours. Specifically, the results  on Rewardbench are:
>
> | Method | Llama 3 8B Instruct | Qwen 1.5 7B Instruct |
> | - | - | - |
> |Vanilla  | 64.8 | 60.9 |
> | PINE| **66.7** | **61.5** |
> | Permutation | 65.9 | 61.3 |
>
> > The variance
>
> Thanks for pointing out this discussion, and sorry for not fully understanding your meaning previously. We follow your suggestion and re-compute the variance of vanilla inference under different shuffle.
>
> Since RewardBench assigns different weights to different subsets, and our originally dumped results do not record the belonging of each sample, we will use an equal average to deliver the results here (therefore, you may find the results are not the same as the Paper reported). Results still show that PINE is effective when all samples are equally averaging.
>
> Llama 3 Series:
>
> | Method |8B | 70 B |
> | - | - | - |
> |Vanilla (GT at A)  | 70.7 | 80.1 |
> |Vanilla (GT at B)   | 65.5 | 76.5 |
> |Vanilla (Shuffle)  | 68.1 $\pm$ 0.4 | 78.3 $\pm$ 0.3 |
> | PINE| **70.5** | **81.5** |
>
> Qwen 1.5 Series:
>
> | Method | 1.8B |  4B | 7B | 32B | 72B | 72B (Qwen 2.5) | 110B |
> | - | - | - | - | - | - | -| -|
> |Vanilla (GT at A)  | 39.1 | 32.6 | 60.9 | 75.2 | 80.3 | 90.0 | 88.0 |
> |Vanilla (GT at B)   | 63.1 | 72.2 | 57.2 | 74.5 | 69.0 | 82.5  | 74.7 |
> |Vanilla (Shuffle)  | 51.1 $\pm$ 0.5 | 52.4 $\pm$ 0.6 | 59.1 $\pm$ 0.5 | 74.8 $\pm$ 0.4 |  **74.6** $\pm$ 0.4 | 86.2 $\pm$ 0.3 | 81.3 $\pm$ 0.4 |
> | PINE| **55.4** | **56.6** | **63.0** | **78.2** | 74.3 | **87.5** |**85.0** |

---

> > ### Author Response · Authors · 2024-12-02
> >
> > Thank you for engaging with us and giving insightful feedback!
> >
> > Since the rebuttal is going to end, please let us know if you have any additional questions and we hope our replies address your concerns.
> >
> > If you find our responses helpful, we would greatly appreciate it if you could consider raising your scores.

---

> > > ### Comment · Reviewer_Qpxn · 2024-12-03
> > >
> > > Thank you for your response. I am now more confident that your work improves over the state of the art. As a result, I have raised my score. Good luck!

---

> > > > ### Author Response · Authors · 2024-12-03
> > > >
> > > > Thank you for your valuable feedback, and engage with us in the rebuttal phase!
> > > >
> > > > We will incorporate your advice into our next revision to make the paper more convincing.

---

### Official Review · Reviewer_K8Th · 2024-11-06

**Soundness:** 2
**Presentation:** 3
**Contribution:** 3
**Rating:** 8
**Confidence:** 4

**Summary:**

Positional bias is a major problem for language models. Several proposals have been made in literature towards mitigating this issue. This paper proposes one method by making the inter-document attention bidirectional. Specifically, they propose their own order such that the effects of order dependence in positional encoding is minimized.

**Strengths:**

- I like this paper. they propose a bidirectional attention in the input documents to achieve input order independent attention.
- Their results are solid. For order dependent benchmarks, they do achieve significant gains.
- While the paper is solid, I would like the authors to revise their claims of position invariance to input-position invariance.

**Weaknesses:**

- I tried understanding Figure 2 many times, it was not clear to me. Please see question below. The only sensible explanation is that the attention is simulating the document under question being placed at the end of the sequence, with nearness being provided by importance scores.
- What about the n^2 importance score computations per set of n documents. How to do this when there are multiple documents?
- The position invariance is quite misleading. The correct term I believe would be importance dependent position weighting. i.e. You know the documents far away to the current document under consideration have higher likelihood of error, and hence you place them far away, such that the impact of position is mitigated.
- Here is a counterexample for position invariance: Let us take a query which involves multi-hop retrieval. The query asks: Retrieve the home pages of all the professors working in the CS department of X university, and list out their topics of interest. Here the retrievals may be all nearly equally important. A full position invariance would require an independence with order whereas that cannot be achieved through this method. What you instead do is to rank documents by importance order and then mitigate positional dependence by placing them in that order of closeness for attention. This is not "invariance". My major concern is with the nomenclature, which claims invariance, and eliminating position bias.
- Lemma 1 may need a revision: sorting by computed importance scores introduces an implicit positional bias. While the function \( f(\text{input order}, \text{content}) \rightarrow \text{output permutation order} \) maintains input order invariance if \( f(\text{io1}, \text{content}) = f(\text{io2}, \text{content}) \) for all input orders \(\text{io1}, \text{io2}\), it imposes a new order-dependent bias in the output, meaning true position invariance is not achieved. Thus, while input-position invariance is maintained, true positional invariance is not achieved, as the sorting process introduces a new, order-dependent bias in the output.

**Questions:**

- Figure 2, I understood the last row. 8 is at its original place and (4,5)> (6,7) > (2,3).
    - Rows 6 and 7 -- (6,7) is at its original place and (2,3) > (4,5)
    - Rows 4 and 5 -- (4,5) should be at their original place, but (6,7) is! Why revise the position of this document to the end when both the documents can be (equally) at small distance from the document under consideration?
- I only understood Figure 2 after going through paragraph 2 in lemma 1. Please consider revising the write up with the document order for computation of each of the rows of the attention.

### Suggestions:
- Move paragraph 2 from Lemma 1 proof from appendix to main paper to explain your method.
- Consider revising the words position invariant to input-position invariant. Please also consider revising the claim that the position bias of LLMs is eliminated (in the title of the paper). From my best understanding, it is mitigated through a reordering and not eliminated.

---

> ### Author Response · Authors · 2024-11-22
> **Rebuttal (1/1)**
>
> We thank you for your insightful advice and suggestions! We are glad that you like our paper and find our results solid. We answer your questions below:
>
> > W1 & Q1: Figure 2 Understanding
>
> Your overall understanding is correct: important documents are placed near to the end of the sequence in PINE.
>
> For your understanding of row 4,5, your suggestion is a better choice than ours if models are not trained bi-directional (e.g., Prefix LM instead of causal LM). Unfortunately, most modern LMs are trained casually, which means the query token can only see previous tokens. Document 2 is the query document in rows 4,5; suppose its position is (4,5), and Document 3 has the position (6,7). Then, attention computation will make query document 2  see “future” document 3, which is not seen and trained by LMs and causes poor performance.
>
> > W2: Importance score computation
>
> This is the same normal attention computation and therefore can be computed in parallel. We just follow the pipeline of attention score, with the only modification being that Q and K are not encoded with position embedding.
>
> > W3 & W4 & W5: About our claim, proof and terminology
>
> We believe both ours and your understanding is correct. The difference is that we stand on different perspectives. We say “position-invariant” from the input-output perspective: results remain unchanged regardless of input orders. Your proposed claim stands on the method implementation perspective.
>
> The “elimination” also originates from the input-output perspective. Our method still uses position encoding, and we find that removing position encoding such as NoPE yields poor performance in our preliminary experiment.
>
> We agree to add your understanding to the main body of the paper and clarify more about “elimination” and “invariance” in our paper. Please see Section 3.3 green texts.
>
>
> > Q1 & S1 & S2: presentation and terminology improvement
>
> We thank you for your suggestion and moved the Lemma 1 to our main body. As our response to your W3-5 promises, we’ve added clarification on the terminology on “elimination” and “invariant” is from an input-output perspective and address that internally, the position encoding is still used. Please see Section 3.3 green texts.

---

> ### Comment · Reviewer_K8Th · 2024-11-22
> **Inc of score**
>
> Thanks authors! I was already in favour of the paper, but improving "position-invariance" terminology to eliminating bias, clarifies my major concern.
>
> Increasing my score. Good luck!

---

> > ### Author Response · Authors · 2024-11-25
> > **Response to Reviewer K8Th**
> >
> > Thank you for your helpful advice and for reading our rebuttal! We are glad that our responses address your concerns.
> >
> > We will officially incorporate your suggestions into our paper's next revision.

---

### Official Review · Reviewer_azrE · 2024-11-06

**Soundness:** 2
**Presentation:** 1
**Contribution:** 3
**Rating:** 5
**Confidence:** 3

**Summary:**

The authors introduce a method to de-bias a positional dependence for concatenated documents in the context of a LM and prove its supremacy against other de-biasing baselines.

They also compare against vanilla inference, but I cannot tell whether the numbers are actually beneficial.

**Strengths:**

- very relevant problem
- the recency and primacy biases are interesting.

**Weaknesses:**

- Figure 2 is confusing, since the documents change color when you move them (in the right plot). So it looks like the $D_3$ in the first row is the tokens [2,3], but in the second assignment iteration suddenly $D_3$ is [4,5]? It seems inconsistent with the last row, where importance ordering is $D_2 > D_3 > D_1$, but they are not ordered in that way.
- I don’t think you can claim that you pinpointed the causes of positional bias in transformers. Everybody knows what those are.
- Please Correct me if I am wrong here.
The proof in App B shows that the softmax term is invariant under inter-document permutation before the ordering takes place, but Lemma 1 takes the QKV invariance as a given.
How I think Lemma 1 and the proof should go:
Lemma 1: “The PINE algorithm makes $H_{Pine}$  document-position invariant”
Proof: “The algorithm sorts the documents by a document-position invariant metric and concatenates them. Therefore, the resulting concatenation  is invariant. $H_{Pine}$ is only a function of that concatenation, therefore $H_{Pine}$ is invariant. Proof ends”.
- I can’t tell if the numbers are good. The method outperforms other methods of debiasing, but it does not seem to do super well against Vanilla (shuffle) in Figure 4 for instance. Given that you shuffle reward bench, is Vanilla in Table 2 also implicitly Vanilla (shuffle)? It is overall confusing to me, but I admit that this is an opinion.

**Questions:**

- Please discuss the relationship to NoPE (https://arxiv.org/pdf/2305.19466) in your related work
- I find the importance measure that you introduce a little bit weird if you make it independent of positional bias, since you will later calculate attention with positional bias in there, which might change the importance quite a bit. Have you found your measure of document importance, without positional bias, to be monotonic with an importance measure that calculates the attention with positional bias, but putting each document candidate at the same position, right before the currently decoded document for instance?
- I am overall confused by the introduction of bidirectional attention. Given that you calculate importances independently of relative document-position, and you reorder anyway, why do you have to have an attention in the forward direction, instead of simply always putting the document you are decoding at the last position? I might have understood this wrong, the first half of Page 5 is a little confusing to me. When following the concrete example in the proof, there is no bidirectional attention, right?
- Can you comment on the downsides of making keys query dependent (L235)? This seems like you will loose parallelism, at the very least the option to KV-cache.
- Can you comment on your intuition as to why SP and PCW are worse than your approach?
- You say you introduce two other debiasing baselines, permutation and calibration. Is permutation the same as Vanilla (shuffle) in Figure 4? Why are the numbers for it not in any tables?
- Table 2: I don’t fully understand the baselines. IIUC, the GT is either at A or at B, which is why you say 50% random guess baseline, i.e. there is no position C. Why is Vanilla (without qualifier) worse than both Vanilla (GT at A) and (GT at B)? it is not always one of the two, which would lead to some kind of average?
- what do you mean by reasoning “pairs” in 418ff?

---

> ### Author Response · Authors · 2024-11-22
> **Rebuttal (1/2)**
>
> Thanks for your careful review. We are glad that you find our research interesting and relevant. We answer your questions with bullet points:
>
> > W1: Figure 2 is confusing
>
> Numbers in Figure 2 are positions when tokens serve as **keys**, and the numbers of the diagonal are also positions when tokens serve as **queries**. The importance score is computed between **queries** and **keys**. Therefore, the ranking of the documents’ importance differs for different queries, and the claim “The importance ordering is D2 > D3 > D1” is incorrect since this is only the case for Token 8. The [2,3] and [4,5] correspond to the first two formulas in Figure 2, so they are consistent. We mention all these parts in our paragraph of line 219 and 226.
>
> > W2: pinpointing the causes
>
> We aim to emphasize that they are the **only** two parts that cause position bias, which previously no people explicitly discussed to the best of our knowledge. This discussion offers the necessary background to explain why we only focus on these two parts. We do not address this as our core contribution. In the revised paper, we remove words like “our” and use “revisit” in the abstract and the end of the introduction and highlight the modification in the green color.
>
> > W3: Alternative proof
>
> Both yours and ours are correct understanding. The core of Lemma 1 is to say $H\_{pine}$ does not include any new position information, regardless of whether inputs have it. We let input QKV be position-invariant in Lemma 1 to address the premise of mathematical induction, which is used in the proof of theorem 1.
>
>
> > W4: The results
>
> Yes, Vanilla means Vanilla(shuffle) in Table 2.  We apologize for the confusion and have changed the annotation accordingly. PINE performs similarly to vanilla inference in Figure 4, which we discuss in line 479 and we hypothesize the reason is that ordering becomes difficult for LMs when there are more documents. However, our results show that PINE consistently performs better in RewardBench, molecule generation, and math reasoning.
>
>
> > Q1:Discuss NoPE
>
> Thank you for the suggestion. We’ve added it to the updated version to Section 2 (highlighted with green color). NoPE removes the positional encoding, which we find the results are very low in our pilot experiments.
>
>
> > Q2: The importance measure
>
> We hope we understand your question correctly: Will the importance ranking change before and after applying position encoding? The answer is “sometimes”. The ideal solution is no longer applying position encoding (i.e., NoPE). However, this solution leads to much worse results in our preliminary experiments. Therefore, we keep positional encoding and try to make it not affect the importance ranking: re-order descendingly since positional encoding has recency bias. In this way, RoPE mostly respects the ranking order. Nevertheless, RoPE has an oscillation feature at the microscopic level so the importance ranking is not guaranteed unaffected. This phenomenon does not affect our proof of “elimination.”
>
>
> > Q3: The bidirectional attention
>
> As shown in Figure 2, the reorder determines the position of documents, and bidirectional attention only applies to query documents (i.e., for token 8, the attention is causal), which lets a query document “see” all other documents when computing attention score. In the proof, bidirectional attention is still used since when computing H_i, query document i need to see all documents.
>
> > Q4: the dependence of keys and queries
>
> This does not affect the KV cache. The conventional KV cache is to cache KV after applying RoPE. In our implementation, we cache KV without RoPE and apply RoPE when needed. In the inference, the bottleneck is IO (loading from HBM to SRAM) instead of compute, and RoPE only brings lightweight computation. Therefore, PINE does not affect KV cache effectiveness and efficiency. The downside is that the implementation requires more engineering, and since we are not experts in writing efficient codes, we still keep the “for” loop in the implementation, which makes the real wall time slower.
>
> > Q5: About PCW and SP
>
> We hypothesize that PCW and SP introduce more OOD operations. For example, PCW and SP assign different tokens to the same positions. However, LM is not trained to handle such operations. Our method will always assign different tokens to different positions when serving as keys in the “eye” of each query. Another reason is that they lose contextual information, while PINE keeps it.

---

> > ### Author Response · Authors · 2024-11-22
> > **Rebuttal (2/2)**
> >
> > > Q6: About Permutation Baseline
> >
> > We apologize for not making this obvious. We did not include this in tables due to space limits, and you can find results in the L459 paragraph. In short, the calibration-based method has non-sense outputs, which we believe is caused by its strong assumption. Permutation-based methods perform worse than ours. We’ve highlighted permutation results in our updated version by adding a bolded sentence at the beginning of the paragraph (L459), and we highlight them with a green color for you to locate.
> >
> > > Q7: About Vanilla (Shuffle) performance v.s. Vanilla (Gt at A and B)
> >
> > This is a binary classification problem, but Vanilla (Shuffle) is not necessarily the average of two extreme cases. It could perform even worse than a random guess if the model capability is limited. The key problem is that the model may prefer different positions for different problems. For example, if the dataset has ten questions and the model prefers option A the first five questions and B for the rest, then GT at A and GT at B will all have 50% acc. However, Vanilla(shuffle) may have up to 100% and low to 0%, depending on the concrete shuffle.
> >
> >
> > > Q8: The meaning of reasoning pairs.
> >
> > We apologize for the confusion. This is a typo and should be “reasoning problems in RewardBench.” We’ve corrected this in the updated version in L424.
> >
> >
> > Overall, we sincerely thank you for your suggestions. We are happy to modify the paper further if you have any suggestions for making the presentation more precise. Please let us know if you have follow-up questions, and we will be glad to discuss.

---

> > > ### Comment · Reviewer_azrE · 2024-11-22
> > >
> > > Thank you for your clarifications. Some things became clearer.
> > >
> > > I maintain my position on the proof, it can be put into a few lines, since it is rather trivial. However, as another reviewer suggested, the example you walk through in the proof can be used to further clarify the method.
> > >
> > > Figure 2 remains confusing to me despite your attempts at clarification. Can you explain why there are white boxes on every second line after on the one-off diagonal? Maybe you could write down the resulting attention expressions for one or two of the lines in this figure.
> > >
> > > Maybe it helps to compare against a method that seems close, that I have understood: When reading more carefully through the related works, I noticed Peysakhovich & Lerer (2023) also sorts inputs according to their importance. Are you comparing against it? That baseline seems quite relevant and shows good improvements in the paper. If I understand correctly, the biggest difference between this and your work is that they calculate attention __with__ inter-document position information, and then they sometimes re-sort to account for that. It looks like that re-sorting can have a big effect, which relates to my initial Q2.

---

> > > > ### Author Response · Authors · 2024-11-23
> > > > **Follow-up responses**
> > > >
> > > > We thank you for your prompt reply. We clarify your further questions below.
> > > >
> > > > > The proof
> > > >
> > > > We agree that the proof can be relatively simple. The proof actually does not necessarily need to be written as it is a direct application of the symmetry principle, which is broadly used in natural sciences such as Physics. However, we think the CS community may be unfamiliar with symmetry principle so we write a detailed proof with examples instead.
> > > >
> > > > > White boxes in our attention
> > > >
> > > > As discussed in L213-L215, we use bidirectional attention across documents and causal attention inside each document. The white box is because it is inside one document.
> > > >
> > > > For line 4, the query is the first token of the second document, and it should see document 1, document 3, and itself when computing attention matrix.
> > > >
> > > > For line 5, the query is the second token of the second document, and it should see document 1, document 3, the first token of document 2, and itself when computing the attention matrix.
> > > >
> > > > > Comparision with Peysakhovich & Lerer (2023)
> > > >
> > > > First, we hope to point out that Peysakhovich & Lerer (2023) cannot eliminate the position bias since their method does not follow the symmetry principle. Another difference besides what you point out is that we use bidirectional attention, whereas they use causal attention as usual.
> > > >
> > > > Second, our method includes re-sorting, too. The difference is that we only do one re-sorting instead of periodically re-sorting. Figure (2) numbers are the results of re-sorting, which we discuss in L226. Figure 4(b) also discusses several variations of re-sorting techniques and shows that ours performs best, which addresses the effectiveness of re-sorting.
> > > >
> > > > Third, that paper does not release codes and lacks some details for us to implement ourselves. For example, how is the document attention normalized? Is the process running for every decoding step or just once? Therefore, we can not faithfully reproduce the results.
> > > >
> > > > Lastly, we want to re-address your Q2. Peysakhovich & Lerer (2023) could fall into local optima due to position encoding, and our importance ranking may be affected by RoPE's local oscillation nature (See RoFormer https://arxiv.org/pdf/2104.09864 Figure 2 for better understanding). Neither of these methods is perfect, and we both show the importance of re-sorting in our paper (Figure 4 b of ours and Figure 5 of Peysakhovich & Lerer (2023) ), though we use different re-sorting techniques.
> > > >
> > > >
> > > > We hope our responses make things clearer and thanks again for reading our rebuttal.

---

> > > > > ### Comment · Reviewer_azrE · 2024-11-24
> > > > >
> > > > > The CS community is most certainly familiar with symmetry.
> > > > >
> > > > > Thanks for the clarification about the white boxes, that helped. I think I also understood the numbers now, and this effectively comes down to re-sorting to calculate the attention for each token (although that would of course be a bad implementation)
> > > > >
> > > > > Peysakhovich & Lerer (2023) are not far from eliminating position bias, they only need to calculate their sorting metric without RoPE like you do.
> > > > > Then, the remaining difference to your method is the bidirectional attention. Did you ablate what kind of difference that makes? Apologies, I should have asked this in the first review, but I have just recently realized how close Peysakhovich & Lerer (2023) your method is.

---

> ### Author Response · Authors · 2024-11-25
>
> Thanks for reading our rebuttal and your quick reply.
>
> > About the symmetry principle
>
> To the best of our knowledge, this term is not frequently used in CS papers as in Physics papers, so we have to be conservative when writing proofs. As a preliminary statistical evidence, only 0.5% (3479 out of 652,604) CS papers in arxiv mention "symmetry", whereas the portion is 7.9% (112,784 out of 1,432,095) in Physics papers.
>
> However, we are more than happy to add a one-sentence illustration about the symmetry principle to our paper after we agree on other points (just to avoid back-and-forth revision).
>
> > Peysakhovich & Lerer (2023) are not far from eliminating position bias, they only need to calculate their sorting metric without RoPE like you do.
>
> No, they can not eliminate position bias even if they calculate without RoPE. The causal attention breaks the symmetry.
>
> > Is bidirectional useful?
>
> Of course, the red dashed line in Figure 4(b) outperforms the blue dashed line with ~1% Accuracy, showing that simply adding bidirectional attention without our re-sorting (and therefore, the computation overhead of re-sorting is discarded) is useful.
>
> > but I have just recently realized how close Peysakhovich & Lerer (2023) your method is.
>
> Thank you for your question. We hope to address our differences:
>
> * We use bidirectional attention
>
> * Because of the bidirectional attention, our re-sorting is different from their re-sorting, we need to design a position assignment strategy after re-sorting in accordance with our bidirectional attention.
>
> * We need to exclude RoPE when computing importance scores.
>
> * We do not need to periodically re-sort to find a fixed point.
>
> * We have theoretical guarantees.
>
> To sum up, although both methods use "re-sorting," the re-sorting itself and the ways to incorporate the re-sorting results differ greatly.
>
> Thanks again for your feedback; we hope we address your concerns and questions.

---

> > ### Comment · Reviewer_azrE · 2024-11-26
> >
> > Re bidirectionality:
> > I don't see the blue dashed line in Figure 4b. Do you refer to the blue dashed line in 4a?
> >
> > Re position invariance:
> > I want to decode a new token from my current document. Now I look at all other documents (they are in my context) and calculate a aggregated importance measure that does not depend on their respective positions (i.e. without rope).
> > Now, I define position invariance (document-wise): f(set of documents D, order σ_j) = f(D, σ_i) for all permutations σ_i,σ_j.
> > If f includes a sorting S (depends on D implicitly) via position invariant metric (since rope not included), then for all σ_i, f(D, σ_i) = g(D, S(σ_i)) = g(D, σ_0) = g(D, S(σ_0)) = f(D, σ_0). σ_0 is the sorted permutation. I am confused, what am I missing?
> >
> >
> > The baseline of Peysakhovich & Lerer (2023) is still important to compare against I believe.

---

> ### Author Response · Authors · 2024-11-27
>
> Thanks again for your quick reply. We are delighted you are engaging with us frequently!
>
> > Blue line
>
> Yes, and they have ~1% Acc difference.
>
> > Position invariance
>
> Your most understanding is correct. The minor mistake is "calculate an aggregated importance measure that does not depend on their respective positions (i.e. without rope)." Causal attention implicitly contains position information; therefore, the importance score still depends on positions if causal attention is used instead of bidirectional attention or PCW (masked-out inter-document attention). All other parts you mentioned are correct.
>
> Therefore, that's why we need bidirectional attention (PCW does not show good performance according to our experiments reported in our paper)
>
>
> > The baseline of Peysakhovich & Lerer (2023) is still important to compare against I believe.
>
> Yes, we agree this baseline is important. However, the paper does not release code and we cannot obtain results.
>
>
>
> Again, we thank you for your valuable feedback and hope our response could clarify your questions.

---

> ### Author Response · Authors · 2024-12-02
> **Additon experiments on Peysakhovich & Lerer (2023)**
>
> We implement the method ourselves to show the difference between PINE and Peysakhovich & Lerer (2023), and find PINE still get a better result on RewardBench.
>
> On LLama 3 8B Instruct:
>
> | Method |  Accuracy |
> | - | - |
> | Vanilla (Shuffle) | 64.8|
> | Peysakhovich & Lerer (2023) [k=1] | 65.2 |
> | PINE | **66.7** |
>
> We believe our additional experiments can help you better understand the differences between the two methods.
>
> Since the rebuttal is going to end, please let us know if you have any additional questions and we hope our replies address your concerns. If you find our responses helpful, we would greatly appreciate it if you could consider raising your scores.

---

> > ### Comment · Reviewer_azrE · 2024-12-02
> >
> > thanks for the additional experiments. i am raising my score

---

> > > ### Author Response · Authors · 2024-12-02
> > >
> > > Thank you for providing the helpful suggestions! We will incorporate your feedback into our next revision to make the paper clearer and more convincing.

---

### Author Response · Authors · 2024-11-22
**General Rebuttal**

We thank all reviewers for their time and hard work in reading and reviewing our paper. We are happy that reviewers find our method interesting (azrE), useful (1Z1E, 1nRN), have strong results (Qpxn, K8Th, 1nRN), and express favor to our paper (1Z1E, K8Th).

We address reviewers' concerns separately and update our paper to incorporate their suggestions. All modifications are highlighted in green so that reviewers can locate them better. Besides, the main modification is to shorten Section 4.1 under the 10-page limit.

---

### Meta-Review · Area_Chair_Kcpq · 2024-12-18

**Metareview:**

This paper addresses position bias in transformers in situations where generation is supposed to be conditional on a set of input documents.The proposed approach, dubbed PINE, first re-arranges inputs based on relative pairwise document importance scores as measured via aggregate attention scores, which then results in invariant generation given that re-ordering is carried out prior to generation. This comes at the cost of inference overhead since the re-ordering step requires computing importance. PINE operates during inference only and incurs no extra training cost.

The proposal is evaluated in a number of settings generally showing improvements, however it's a bit unclear how inducing position invariance improves so much the performance in reasoning benchmarks such as in the results shown in tables 2 and 3 for RewardBench. If the assumption is that position-dependent models may miss relevant context due to its position, then that could have been verified by exhaustively checking all possible orderings of input documents (perhaps for a subset of the experiments).

The manuscript has some presentation issues, and the method description in section 3.3. as well as the definition of importance scores are a bit confusing and not clear enough. Another limitation worth highlighting is the fact that PINE is limited in scope to settings where input contexts have well defined boundaries, e.g., it is comprised of a set of documents. In a situation where one has a single large document, position bias could still affect performance since relevant information could occur in parts of the document models tend to ignore. Sub-document re-ordering seems to be required in such a case. Moreover, the evaluation, while extensive in a few aspects, has some limitations that are worth noting. For instance, multi-hop settings where information required to answer queries are spread across multiple documents should have been covered.

In summary, the approach offers a strategy to trade compute for order invariance, and consequently boost performance, in settings where generation is conditional on input context and information needs to be retrieved from a set of documents. Although some limitations remain as discussed in the paragraph above, enough evidence is presented to support the authors's claims and the approach would be of value to the community.

**Additional Comments On Reviewer Discussion:**

The main overlapping concerns raised by reviewers revolved around presentation issues and lack of clarity. Authors seem to have addressed those concerns well during the discussion phase, and reviewers were mostly satisfied with the improvements, raising their scores. Although some lack of clarity remains as noted above, and the manuscript would benefit of something like a pseudo-code outlining the inference approach.

---

### Decision · Program_Chairs · 2025-01-22

Accept (Poster)